# Multiplexed histology analyses for the phenotypic and spatial characterization of human innate lymphoid cells

Anna Pascual-Reguant [1,2], Ralf Köhler[2], Ronja Mothes[1,2], Sandy Bauherr [2], Daniela C. Hernández [3], Ralf Uecker[1,2], Karolin Holzwarth[2], Katja Kotsch[4], Maximilian Seidl [5,6], Lars Philipsen [7], Werner Müller[8], Chiara Romagnani [3,9], Raluca Niesner [10,11] & Anja E. Hauser [1,2 ✉]

Innate lymphoid cells (ILCs) emerge in the last few years as important regulators of immune responses and biological processes. Although ILCs are mainly known as tissue-resident cells, their precise localization and interactions with the microenvironment are still unclear. Here we combine a multiplexed immunofluorescence technique and a customized computational, open-source analysis pipeline to unambiguously identify CD127+ ILCs in situ and characterize these cells and their microenvironments. Moreover, we reveal the transcription factor IRF4 as a marker for tonsillar ILC3, and identify conserved stromal landmarks characteristic for ILC localization. We also show that CD127+ ILCs share tissue niches with plasma cells in the tonsil. Our works thus provide a platform for multiparametric histological analysis of ILCs to improve our understanding of ILC biology.

[1] Charité - Universitätsmedizin Berlin, Department of Rheumatology and Clinical Immunology, 10117 Berlin, Germany. [2] Immune Dynamics, Deutsches Rheuma-Forschungszentrum (DRFZ), a Leibniz Institute, Charitéplatz 1, 10117 Berlin, Germany. [3] Innate Immunity, Deutsches Rheuma-Forschungszentrum (DRFZ), a Leibniz Institute, Berlin, Germany. [4] Charité - Universitätsmedizin Berlin, Department for General, Visceral and Vascular Surgery, Berlin, Germany. [5] Institute for Surgical Pathology, Medical Center—University of Freiburg, Freiburg, Germany. [6] Institute of Pathology, Heinrich-Heine University and University Hospital of Düsseldorf, Düsseldorf, Germany. [7] Institute of Molecular and Clinical Immunology, Medical Center, Otto-von—Guericke University Magdeburg, Magdeburg, Germany. [8] Miltenyi Biotec B.V. & Co. KG, Bergisch Gladbach, Germany. [9] Charité - Universitätsmedizin Berlin, Department of Gastroenterology, Infectiology and Rheumatology, Berlin, Germany. [10] Biophysical Analysis, Deutsches Rheuma-Forschungszentrum (DRFZ), a Leibniz Institute, Berlin, Germany. [11] Dynamic and Functional in vivo Imaging, Veterinary Medicine, Freie Universität Berlin, Berlin, Germany. ✉email: anja.hauser-hankeln@charite.de

Innate lymphoid cells (ILCs) constitute a heterogeneous and rare cell population of lymphocytes identified only about a decade ago[1]. ILCs were described as the innate counterparts of T cells because although lacking antigen-specific receptors, they share lineage-defining transcription factors (TF), chemokine receptors, and cytokine profiles with T lymphocytes[2]. Since their discovery, ILCs have been extensively studied in several mice and human tissues by flow cytometry and, more recently, by single-cell approaches based on RNA sequencing[3–5] and mass cytometry[6,7]. These added valuable and previously unappreciated information on ILC functional classification, activation states, and developmental trajectories. ILCs are nowadays dissected into cytotoxic ILCs, namely NK cells, and helper ILCs, which are further classified into ILC1, ILC2, and ILC3. While NK cells are CD127$^{-/lo}$, expression of CD127 is a hallmark of all helper ILCs, which are commonly defined as CD45$^+$CD127$^+$ lymphocytes that lack expression of other lineage (Lin) markers, such as CD3, CD19, CD14, CD123, CD141, and FcεRIα [2]. This lineage marker combination is particularly used in humans, with some variations for analyses in mice. Consequently, at least eight markers are needed to unambiguously identify these cells and several more are desired, in order to properly characterize their phenotype. Helper ILCs are mainly known to be tissue-resident cells[8], to be particularly abundant in barrier sites, and to act as sensors for tissue integrity, both promoting inflammation or tissue regeneration and wound healing[1,9–11]. In line with these functional characteristics, their microanatomical localization within and across tissues should fulfill particular needs and, therefore, histological approaches are needed to further understand ILC biology in context, as recent reports have pointed out[12,13]. This requirement holds true not only for ILCs, but for various immune cell subsets, as tissue-associated features generally define tissue and organ function and, thus, decide over health or disease. Whereas conventional immunofluorescence microscopy, due to its spectral resolution, is limited in the number of markers it is able to distinguish within one sample, the ongoing development of multiplexed histology techniques[14–17] addressed and solved this challenge for various biological questions. However, we are not aware of any attempt to use such approaches for the in situ characterization of ILCs and their microenvironment.

In order to map the phenotype and localization of ILCs, we use here multi-epitope ligand cartography (MELC), an automated multiplex microscopy technique that allows high-throughput histological studies. Image data acquisition is combined with a customized computational analysis pipeline. This allows us to unambiguously identify CD127$^+$ ILCs in situ, while further characterizing these cells and their microenvironments, and defining potential microanatomical and molecular fingerprints characteristic for ILC localization and function, in various chronically inflamed tissues. The workflow used for the analysis of high-dimensional image data can easily be adapted for the in situ identification and in-depth characterization of several immune cell types, as well as for mapping particular tissue niches.

## Results

### Generation of highly multiplexed fluorescence-based histology of human samples for the analysis of ILCs.
In order to properly characterize ILCs and their putative hematopoietic and stromal interaction partners in human tissues by immunofluorescence histology, a number of fluorophore-conjugated, easily photobleachable antibodies covering all major leukocyte subsets, as well as stromal markers were tested. In short, repetitive incubation—imaging—bleaching cycles with directly coupled fluorescence antibodies were performed by a pipetting robot, in order to generate a stack of 53 images from the same field of view (FOV).

Each of the images contained information on the pattern and level of expression for a particular marker, at subcellular resolution (325 nm laterally and above 5 μm axially) and at the tissue level, as shown for a tonsil sample in Fig. 1.

### Computational identification and quantification of CD127$^+$ ILCs and other relevant immune and stromal cell types.
As a pre-processing step for image analysis, the acquired fluorescence images were registered and normalized (Supplementary Fig. 1). Afterwards, images were segmented into single cells defined as adjacent clusters of pixels containing both a nucleus and surrounding membrane. For that, we first stacked the images of all extracellular markers and summed them up in order to have one single image containing the spatial information of all membranes (Fig. 2a). Subsequently, a random forest algorithm (Ilastik)[18] used this sum membranes image and the nuclear staining image (DAPI) to perform pixel classification (Fig. 2b, c). In this way, we generated probability maps for nuclei, membranes, and extracellular matrix (ECM) (Fig. 2d) that were subsequently segmented in CellProfiler (CP)[19], to identify single nuclei and single cells. We generated corresponding object masks for nuclei (primary objects) and cells (secondary objects) and applied these masks on the registered fluorescence images for all the markers included in the MELC run, to quantify mean fluorescence intensities (MFI) at the single-cell level (Fig. 3a, b). Subsequently, similar to conventional gating strategies performed in flow cytometry analysis, we set intensity thresholds by visual inspection of the images in order to distinguish negative and positive cell populations for each marker and thereby classify cell-types. Thus, ILCs were defined as Lin$^-$CD45$^+$CD127$^+$ cells, where Lin includes CD3, CD19, CD20, CD14, CD123, CD141, and FcεRIα (Fig. 3b). We used the same fluorescence thresholding strategy to identify other immune cell populations: B cells (CD45$^+$CD19$^+$), plasma cells (CD138$^+$), T helper cells (CD45$^+$CD3$^+$CD4$^+$), cytotoxic T cells (CD45$^+$CD3$^+$CD8$^+$), myeloid cells (CD11c$^+$ and/or CD14$^+$ and/or CD56$^-$CD16$^+$ and/or CD141$^+$) and endothelial cells (CD31$^+$). Cells not matching the aforementioned criteria were defined as others and were not further analyzed. Smooth muscle actin (SMA)$^+$ fibroblasts and stroma-derived ECM structures, such as fibronectin fibers were also identified. With the same computational analysis pipeline, data sets from five independent experiments were examined, the corresponding data tables containing single-cell information with cell-types assigned were generated (Supplementary Table 1) and cells were quantified (Fig. 3c). A total of 6391 ± 350 cells were identified throughout the five data sets. B cells and T helper cells outnumbered the other immune cell populations by a factor of 2–10. While B cells, T cells, and myeloid cells represented rather large cell populations, each accounting for 10–30% of total cell numbers, plasma cells and, especially, CD127$^+$ ILCs were found in very low numbers (Figs. 3c and 4c). The latter only represented 0.15–0.50% of the cells in the analyzed tonsil areas.

Thus, we could accurately identify and quantify rare Lin$^-$CD45$^+$CD127$^+$ ILCs in situ, as well as other relevant immune and stromal cell populations, via intensity thresholding of immunofluorescence images, by using a customized computational analysis pipeline.

### Tonsillar ILCs and plasma cells share distinct microanatomical localizations, defined by a particular stromal composition.
We then aimed to characterize the microenvironment of the identified ILCs. For that, CD127$^+$ ILCs were used as seed points, from which an area of 10 μm radius, representing the average size of a hematopoietic cell, was defined as the ILC niche (tertiary objects) (Fig. 4a). We displayed and quantified both the major immune

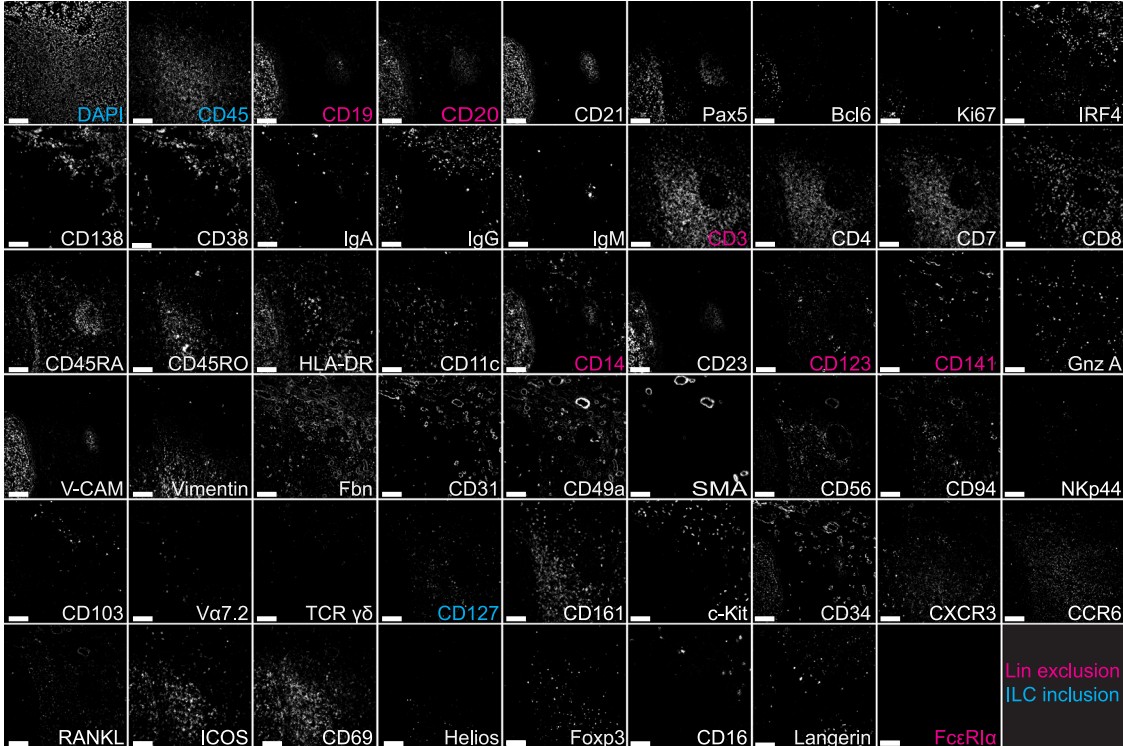

**Fig. 1 Panel overview of a 53 marker MELC run in the human tonsil.** Each image depicts the same field of view, sequentially stained with the depicted fluorescence-labeled antibodies, including surface proteins and transcription factors. The color code highlights markers used for lineage exclusion (magenta) and ILC inclusion (cyan) and markers shown in white represent exploratory markers for ILC and overall cell characterization. Images contain 2048 × 2048 pixels and are generated using an inverted wide-field fluorescence microscope with a 20× objective, a lateral resolution of 325 nm, and an axial resolution above 5 μm. Gnz A granzyme A, Fbn fibronectin, SMA smooth muscle actin, Vα7.2 TCR Vα7.2. Scale bar: 100 μm ($n = 5$).

cell types within ILC niches and important structural components of the tissue, such as ECM proteins and vessels (Fig. 4a, b). The distribution of the various hematopoietic cell populations in those niches showed differences compared to the cell distributions in the whole tonsil areas analyzed (Fig. 4c). In the ILC niches, B cells were of low abundance, accounting only for 10% of the cells and being significantly underrepresented on a per niche basis (Fig. 4b, c). On the contrary, absolute numbers per niche of plasma cells and T helper cells were significantly increased (Fig. 4b). Plasma cells were also significantly enriched within ILC niches and accounted for more than 30% of the cells therein (Fig. 4c). Although T helper cells outnumbered cytotoxic T cells almost 3 to 1 in tonsils, both subsets were equally represented in ILC niches. Myeloid cells were also enriched in the ILC niches (Fig. 4c). Interestingly, 70% of the ILCs analyzed were found within a distance of 10 μm from vessels and almost 80% localized within this distance from the fibronectin fibers (Fig. 4d).

Based on the particular composition observed in the ILC niches and since the tonsil is a highly compartmentalized secondary lymphoid organ, we aimed to also analyze the spatial characteristics of ILCs from a broader tissue perspective. We first identified and characterized in-depth the distinct and well-defined tissue compartments of the tonsils. B cell follicles were defined by extensive CD19, CD20, and CD21 staining that co-localized with the lineage-specific TF Pax5 (Fig. 5a Panel A). The T cell zone was marked by CD3 signal partly co-localizing with other T cell markers, such as CD8, CD4, and Foxp3 (Fig. 5a Panel B). The deep connective tissue septum lining the lymphoepithelial crypts was characterized by SMA and broad fibronectin staining, indicative of fibroblast and stroma abundance, together with enriched CD31 and CD49a endothelial staining (Fig. 5a Panel D). Importantly, plasma cells were almost exclusively localized in this

tissue area, as shown by CD138 staining, which co-localized with other plasma cell-defining markers like the TF IRF4, the membrane marker CD38, and immunoglobulin (Ig) G (Fig. 5a Panel C). We then quantified CD127$^+$ ILCs in each tissue region to evaluate their spatial distribution (Fig. 5b, c and Supplementary Fig. 2). In line with the results obtained for the characterization of the microenvironment of ILCs, we could not detect any CD127$^+$ ILC deep within B cell follicles in the five data sets analyzed, and only ~14% ± 4.8 of CD127$^+$ ILCs were found in the T cell zone. In total, 35% ± 10.6 of the CD127$^+$ ILCs analyzed were located at the border of the B cell follicles, co-localizing with the fibronectin ring that lines such lymphoid structures. Nevertheless, the majority of CD127$^+$ ILCs (64% ± 6.3) were found to be immersed in the subepithelial connective tissue septum and they did not appear isolated, but rather tended to accumulate in groups of 2–4 cells.

Collectively, we could perform an in-depth characterization of the microenvironment of ILCs by using a customized computational analysis pipeline. Both spatial analysis approaches suggest that CD127$^+$ ILCs accumulate in distinct microanatomical areas of the tonsils, where also plasma cells reside, and that are characterized by a high degree of vascularization and a particular stromal composition.

**Clustering analyses of multiplexed histology single-cell data identifies rare tonsillar ILCs.** Since we had generated highly multiparametric single-cell data, we next tested if less biased and time-demanding analysis approaches, not requiring a rather subjective image thresholding, could also be used to identify and characterize ILCs within immunofluorescence images. For that, we used t-distribution stochastic neighbor embedding (t-SNE)

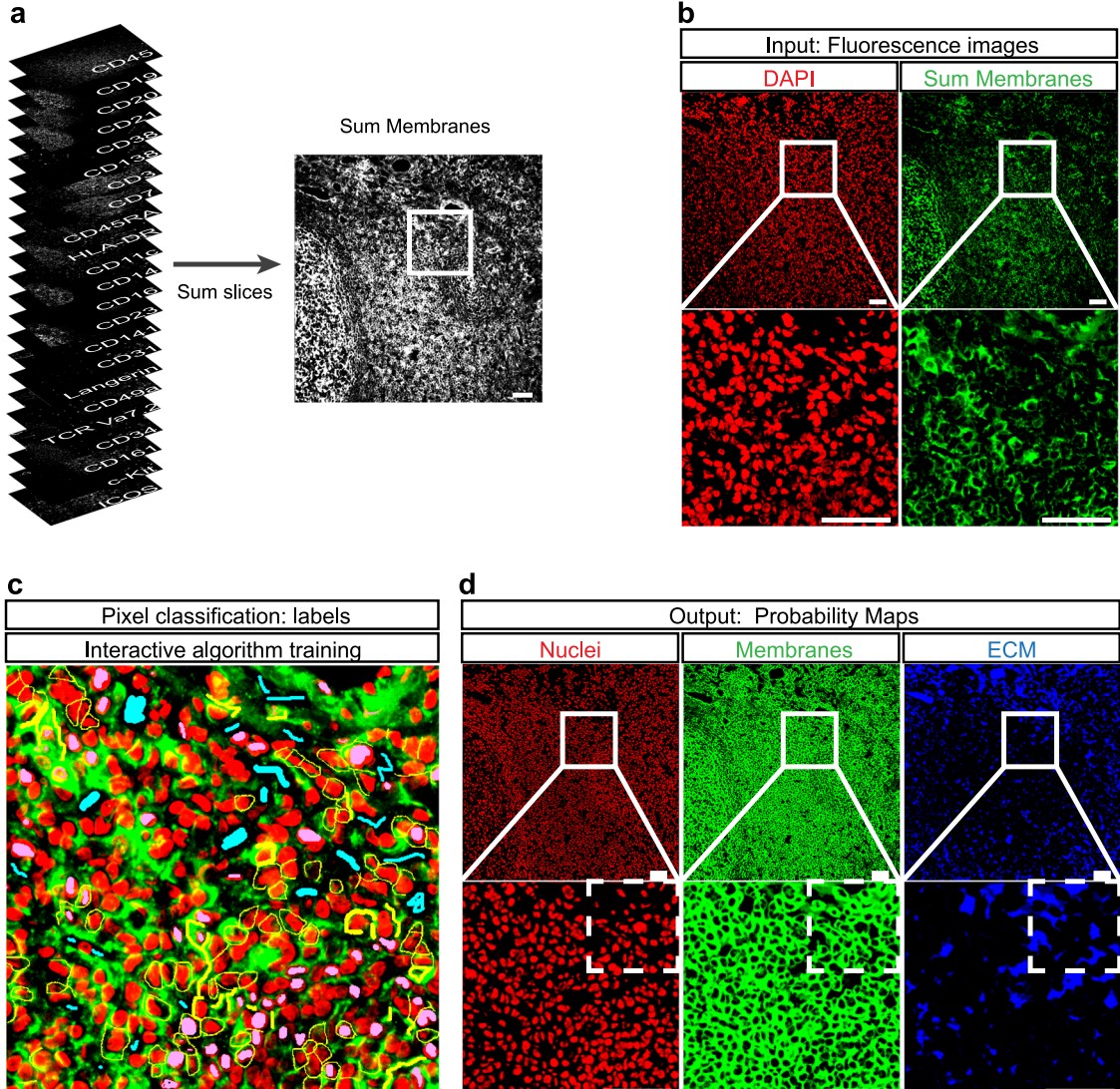

**Fig. 2 Pixel classification by machine learning in Ilastik as a pre-step for cell segmentation. a** A sum membranes image is created by summing of all membrane stainings in ImageJ in order to have one single image containing the spatial information of all cell membranes stained in a MELC run. The white square represents the region of interest (ROI). **b** Fluorescence images from the nuclear staining (DAPI, red) and the sum membranes generated in (**a**) (green) are the input data for Ilastik. Complete images are shown in the upper panel and the ROIs are shown in the lower panel. **c** A machine-learning-based algorithm for pixel classification (random forest) is interactively trained by manually drawing labels for nuclei (pink), membranes (yellow), or extracellular matrix (ECM, cyan) on an ROI of the overlaid input images (nuclear staining in red and sum membranes in green). **d** The trained algorithm calculates probability maps for nuclei, membranes, and ECM in the whole image and in other data sets. The maps can be exported and subsequently used for cell segmentation. Complete probability maps are in the upper panel and ROIs are shown in the lower panel. A second ROI is depicted as dashed lines and represents the tissue area that will be further used as an example in Fig. 3. Scale bars: 50 μm. **a–d** ($n = 7$).

analysis by which a 2-D map was generated, where cells were distributed in nine distinct clusters based on the similarity in expression of all markers simultaneously (Fig. 6a). Cell clusters were annotated based on their protein expression profile illustrated by a heat-map representation (Fig. 6b). Thereby T cells, germinal center B cells, naive B cells, monocytes, macrophages, endothelial cells, fibroblasts, plasma cells, and Lin⁻CD45⁺CD127⁺CD161⁺ ILCs (clustered ILCs) could be identified. We also plotted the pre-classified cells based on our previous fluorescence thresholding approach (pre-classified cells) in the t-SNE map (Fig. 6c) and both cell classification strategies yielded comparable results. Interestingly, 88.5% of clustered ILCs were indeed pre-classified CD127⁺ ILCs, while 7.7% were pre-classified T cells and 3.8% were pre-classified plasma cells. Making use of the spatial resolution of our data, both clustered and pre-classified cells were

plotted in the initial *X*, *Y* coordinate system, resulting in a precise representation of the well-defined tonsil compartments as previously described (Fig. 5a, b and Supplementary Fig. 3). In addition, both clustered and pre-classified ILCs and plasma cells were again shown to preferentially reside in the highly vascularized and fibroblast-rich connective tissue septum (Supplementary Fig. 3).

Thus, we validated the performance of the clustering algorithms on our multiplexed histology data and proved such less biased analysis approaches to be also suited for the identification and spatial characterization of rare ILCs in tissues.

**Clustering analyses of multiplexed histology single-cell data reveals IRF4 expression in tonsillar ILC3s.** We then proceeded to further characterize the identified ILCs by zooming into this

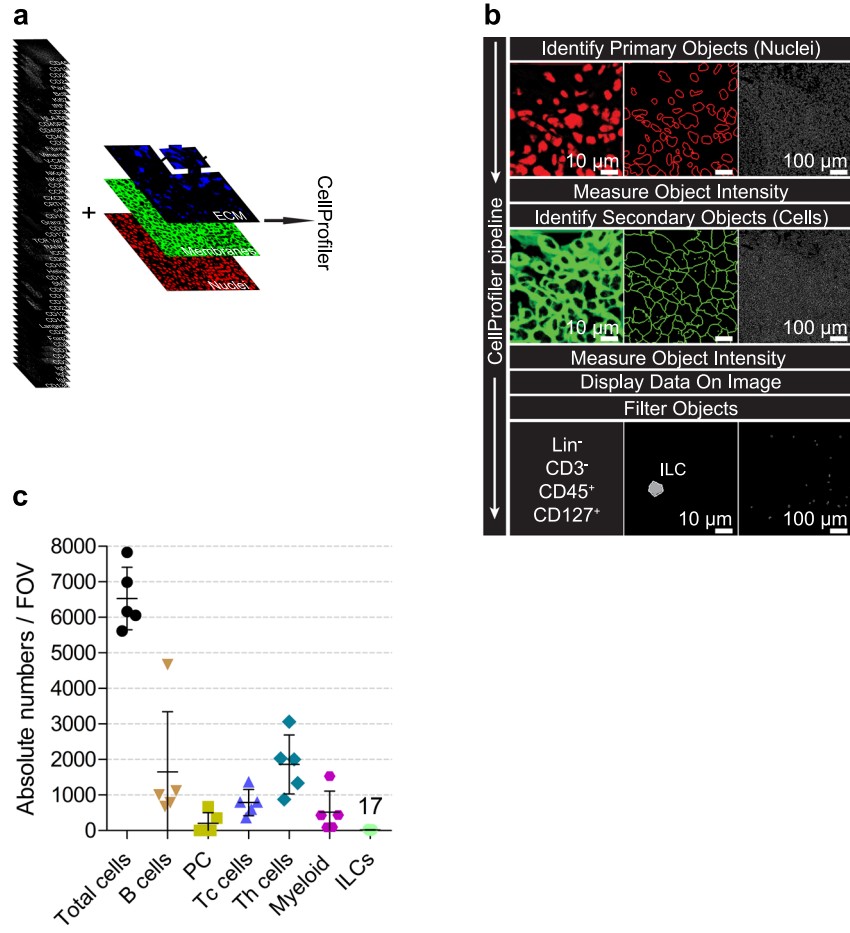

**Fig. 3 Image data analysis allows the identification and quantification of ILCs and other relevant cell populations. a** Fluorescence images and Probability Maps generated in Ilastik are loaded into CellProfiler. The dashed square represents the ROI. **b** Simplified analysis pipeline shows major steps in the data processing. Based on the nuclei probability map (ROI, upper left panel, red), nuclei are segmented as primary objects (upper middle panel: nuclear outlines within the ROI in red; upper left panel: nuclear outlines for the complete image in white). Mean fluorescence intensity (MFI) of nuclear stainings is measured within each nucleus. Using the segmented nuclei as seeding points and together with the membrane probability map (ROI, middle left panel, green), cells are segmented and identified as secondary objects (middle central panel: cell outlines within the ROI in green; middle left panel: cell outlines for the complete image in white). MFIs of membrane stainings are measured in each cell. The display of MFI values per object allows manual thresholding for each marker to classify cells into negative and positive subpopulations. ILCs are identified as Lin$^-$CD3$^-$CD45$^+$CD127$^+$ cells, where Lin includes CD19, CD20, CD14, CD123, CD141, and FcεRIα. Lower middle panel: outline of 1 ILC in the ROI. Lower left panel: outlines of all ILCs in the complete image. **c** Dot plot depicts absolute numbers of relevant immune populations in all tissue areas analyzed. Data are shown as mean ± SD. **a–c** ($n = 5$). Source data are provided as a Source data file.

population and exploring their phenotype. A multiplexed histology-based heat-map of the clustered ILCs showed broad expression of c-Kit together with variable expression of CD7, RANKL, CD69, and NKp44, suggesting an ILC3 phenotype (Fig. 7a), which indeed represents the most conspicuous helper ILC population in the tonsil[6]. Surprisingly, we detected the expression of IRF4 in a subpopulation of ILC3s (Fig. 7a). The TF IRF4 is known to be expressed by B and T cells, supporting their homeostasis by negatively regulating proliferation[20], and IRF4 has also been shown to be essential for the survival and function of plasma cells[21]. However, to date, it has not been described as a marker for ILCs. We first confirmed IRF4 expression on ILCs by visual inspection of the fluorescence images (Fig. 7b). In addition, we aimed to validate our finding at a transcriptional level, and, indeed, the *IRF4* transcript was found to be significantly enriched in sorted tonsillar ILC3s compared to two different hematopoietic progenitor cell populations (Fig. 7c). Importantly, we also confirmed this result at the protein level by flow cytometry, using a standard gating strategy to define the distinct ILC subsets[3,6,22]

(Fig. 7d and Supplementary Fig. 4). 62.7% ± 11.30 of the Lin$^-$CD45$^+$CD127$^+$CD161$^+$ ILCs in the tonsils were found to be IRF4$^+$ and the vast majority of these cells (~98%) were c-Kit$^+$CRTH2$^-$ ILC3s. IRF4 was shown to be co-expressed with both NKp44 and c-Kit (Fig. 7d). In addition, c-Kit$^+$CRTH2$^-$ ILC3s expressed RORγt$^+$, further confirming ILC3 identity (Supplementary Fig. 4). Taken together, the approach used here served us to identify IRF4 as a marker for the phenotypical characterization of tonsillar ILC3s.

Interestingly, the multiplexed histology-based heat-map of the ILC cluster also revealed a subpopulation of ILC3s to be positive for CD138 (Fig. 7a), another plasma cell-related marker, known to bind to ECM and regulate the turnover for soluble factors via its heparan sulfate chains[23]. Since we found plasma cells and ILCs localizing closely together in the tonsils, we needed to rule out signal cross-contamination of adjacent cells due to technical resolution limitations and membrane tiling. Visual inspection of immunofluorescence images confirmed CD138 expression on ILC3s, although at lower levels than on plasma cells (Fig. 7b). In

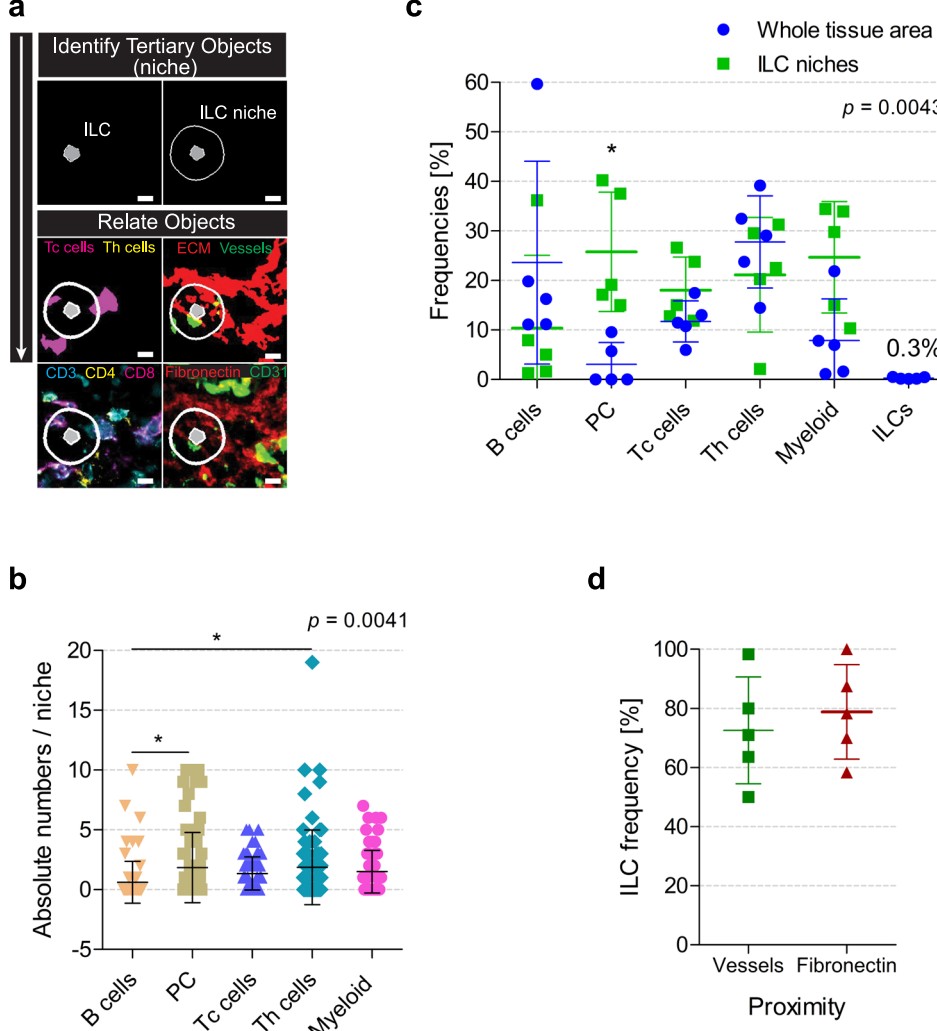

**Fig. 4 Image data analysis reveals a particular hematopoietic and stromal composition for ILC niches. a** In total, 10 μm radius around ILCs is defined as niches. Immune cells and components of the stromal compartment within these areas are quantified. Middle panel: cytotoxic T cells (Tc cells; magenta), T helper cells (Th cells; yellow), vessels (green), and fibronectin fibers (red) within one representative niche. Lower panel: fluorescence images using the same color code. Scale bar: 10 μm. **b** Box plot depicts absolute cell numbers within 74 ILC niches analyzed in five independent experiments. Analysis performed by one-way ANOVA ($F = 3.921$ and 4 *df*) with Bonferroni´s multiple comparisons test, where *$p < 0.01$. **c** Dot plot shows the distribution of cell types in all tissue areas compared to their distribution within ILC niches. Analysis performed by two-way ANOVA ($F = 4.108$ and 4 *df*) with Bonferroni´s multiple comparisons test, where *$p < 0.05$. **d** Box plot shows frequencies of ILCs localized within a 10 μm radius from vessels and fibronectin fibers. **b–d** Data are shown as mean ± SD. **a**, **c**, **d** ($n = 5$). Source data are provided as a Source data file.

addition, an increase in the *SDC1* (*CD138*) transcript was detected in sorted ILC3s, in comparison to hematopoietic progenitor cells (Supplementary Fig. 5). However, we could not detect robust CD138 expression on ILCs by flow cytometry.

**ILCs show distinct localization patterns conserved across tissues.** In order to characterize ILCs in other tissues, we performed MELC in colon samples from two patients diagnosed with ulcerative colitis, using a similar antibody panel like the one previously shown in the tonsil, with minor tissue-specific adaptations (Fig. 8). In this mucosal tissue, as well as in the tonsil, all ILC subpopulations have been well characterized[6,24], but still very little is known about their precise localization. We performed pixel classification of nuclei, membranes, and ECM, segmented both colon data sets, and extracted single-cell MFIs. We plotted all cells expressing at least one of the markers included in the panel in the t-SNE map and nine phenotypically distinct clusters were identified (Fig. 9a). Based on their expression profile

(Fig. 9b), clusters were annotated as HLA-DR[+] T helper cells, T cells, B cells, plasma cells, progenitors, fibroblasts, endothelial cells, and ILCs. ILCs did not express any of the lineage markers and showed high expression of CD45, CD45RO, CCR6, KLRG1, and CD127, indicating ILC identity (Fig. 9b). We further validated the phenotypical characterization of ILCs by visual inspection of the fluorescence images (Fig. 9c). The identified cells were indeed Lin[−]CD45[+]CD127[+] ILCs. While the majority of them were c-Kit[−], CD161[low/-] and Eomes[−], they expressed high levels of CD45RO and variable expression of CXCR3, CCR6, CD69, and ICOS (Fig. 9b, c). We quantified the identified clusters in the two colon data sets analyzed and ILCs accounted for less the 1% of the cells in the tissue (Fig. 10a). Similar to the spatial analysis performed in the tonsil, we compared frequencies of each cluster in the whole tissue with frequencies within ILC niches, which were again defined as an area of 10 μm radius around each ILC. Although the different hematopoietic cell populations were equally represented within ILC niches compared to whole colon areas analyzed, we did find enrichment in fibroblasts in the 10 μm

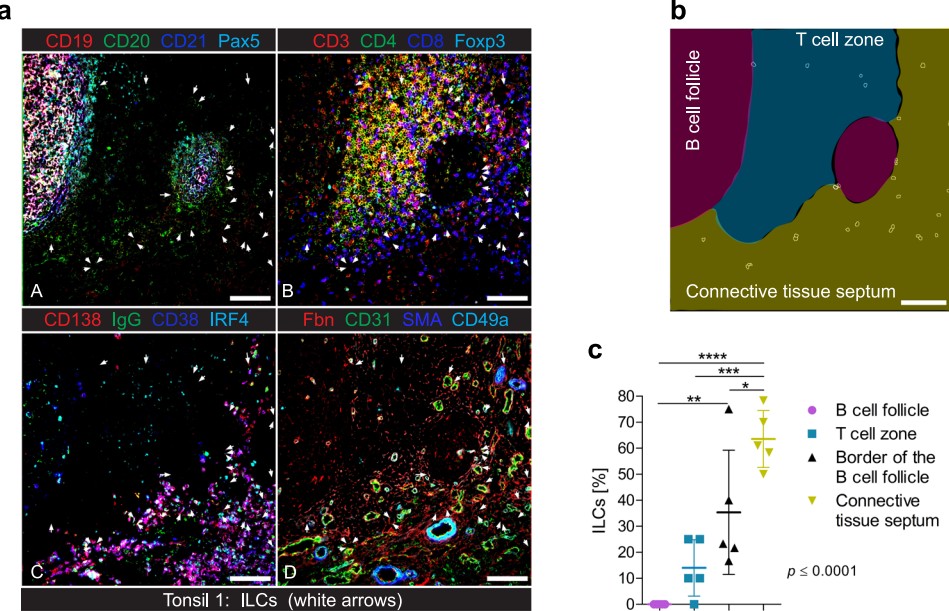

**Fig. 5 ILCs accumulate in distinct microanatomical areas of the tonsils, enriched in the vasculature and connective tissue. a** Overlays of fluorescence images obtained from one single FOV of a tonsil, where ILCs are shown as white arrows. Panel A: CD19 (red), CD20 (green), CD21 (blue), and Pax5 (Cyan) stain for B cells and depict two B cell follicles. Panel B: CD3 (red), CD4 (green), CD8 (blue), and Foxp3 (cyan) stain for T cells and depict the T cell zone. Panel C: CD138 (red), IgG (green), CD38 (blue), and IRF4 (cyan) stain for plasma cells and depict a plasma cell-rich area. Panel D: Fbn (fibronectin, red), CD31 (green), SMA (smooth muscle actin blue), and CD49a (cyan) stain for extracellular matrix (ECM) proteins and endothelial cells, and depict the connective tissue septum. Scale bars: 100 μm. See also Supplementary Fig. 2. **b** Three distinct functional areas in the tonsil (B cell follicles in magenta, T cell zone in blue, and connective tissue septum in yellow) are shown as segmented tissue compartments. Scale bar: 100 μm. **c** Box plot showing the distribution of ILCs within each tonsil compartment, as segmented in (**b**). Data are shown as mean ± SD and analyzed by one-way ANOVA ($F = 19.01$ and 3 $df$) with Bonferroni´s multiple comparison test, where $*p < 0,05$, $**p < 0,01$, $***p < 0,001$ and $****p < 0,0001$. **a–c** ($n = 5$). Source data are provided as a Source Data file.

areas surrounding ILCs. Interestingly, in line with our findings in the tonsils, also colonic ILCs tended to form groups of 2–4 cells in close proximity to fibronectin and collagen fibers, generally coating CD31[+] endothelial cells (Figs. 9c and 10b).

Collectively, we show that the histological and analytical approaches used here are well suited for the identification and characterization of ILCs in several human tissues and provide information on the spatial distribution of ILCs. We show ILCs to be tightly associated with structural components that shape tissue microstructure.

## Discussion

ILCs have emerged in the last few years as key players not only in the regulation of immune responses but also of broader biological processes, such as tissue metabolism, regeneration, and wound healing[25], suggesting tight interactions with their environment. However, despite the wealth of knowledge recently generated in the ILC field, and despite their relevance for tissue homeostasis and repair, neither the precise localization of human ILCs in tissues nor their interactions with the microenvironment have been characterized in depth. This knowledge may be crucial to understand the putative heterogeneity and plasticity of ILCs associated with both health and disease. For that, highly multiplexed histology techniques are needed, allowing for single-cell, spatially resolved omics analysis of solid tissues. Lately, imaging mass cytometry (IMC)[17], CODEX[14] and tissue-based cyclic immunofluorescence (t⁻CyCIF)[15,26] have been developed and successfully coupled to computational analytic pipelines, aiming to evaluate high-dimensional image data[27]. However, the analyses thereby performed so far have mainly focused on the phenotypical characterization of large immune cell populations[14] with

known microanatomical tissue localization[28] and their interactions with the microenvironment in the context of tumor immunology[26,29].

Here we use the multiplex microscopy technique MELC[16] for the in situ characterization of rare ILCs and their microenvironments in human mucosal tissues. The analysis consists of inclusion and exclusion markers needed for ILC identification, a combination of markers used to further characterize ILCs, and several exploratory markers. The latter is meant to give us information on tissue composition in general, allowing us to dissect both the hematopoietic and the stromal compartment.

After image registration and normalization, we performed cell segmentation of the acquired and registered images in a two-step process as suggested by Shapiro et al.[27]. Through machine learning approaches, we could make use of all the information contained in the multiparametric image data set to create enhanced multichannel images that cover the entire nuclear, membrane, and ECM signal. One tonsil data set was sufficient to train the random forest algorithm, as validated by visual inspection, and pixels on the other four data sets were robustly classified without re-training. By using this pixel classification algorithm, we gained accuracy in the consecutive segmentation steps, through which we identified single cells within the images. We classified cells into relevant immune cell types based on fluorescence thresholding of the images from lineage-defining markers and quantified these in five independent experiments performed with tonsil samples.

Total cell numbers per FOV were found to be consistent throughout data sets of the same tissue origin. The variability among experiments in the absolute numbers and frequencies of the various immune cell types reflects the high compartmentalization of the tonsil and the variable representation of each

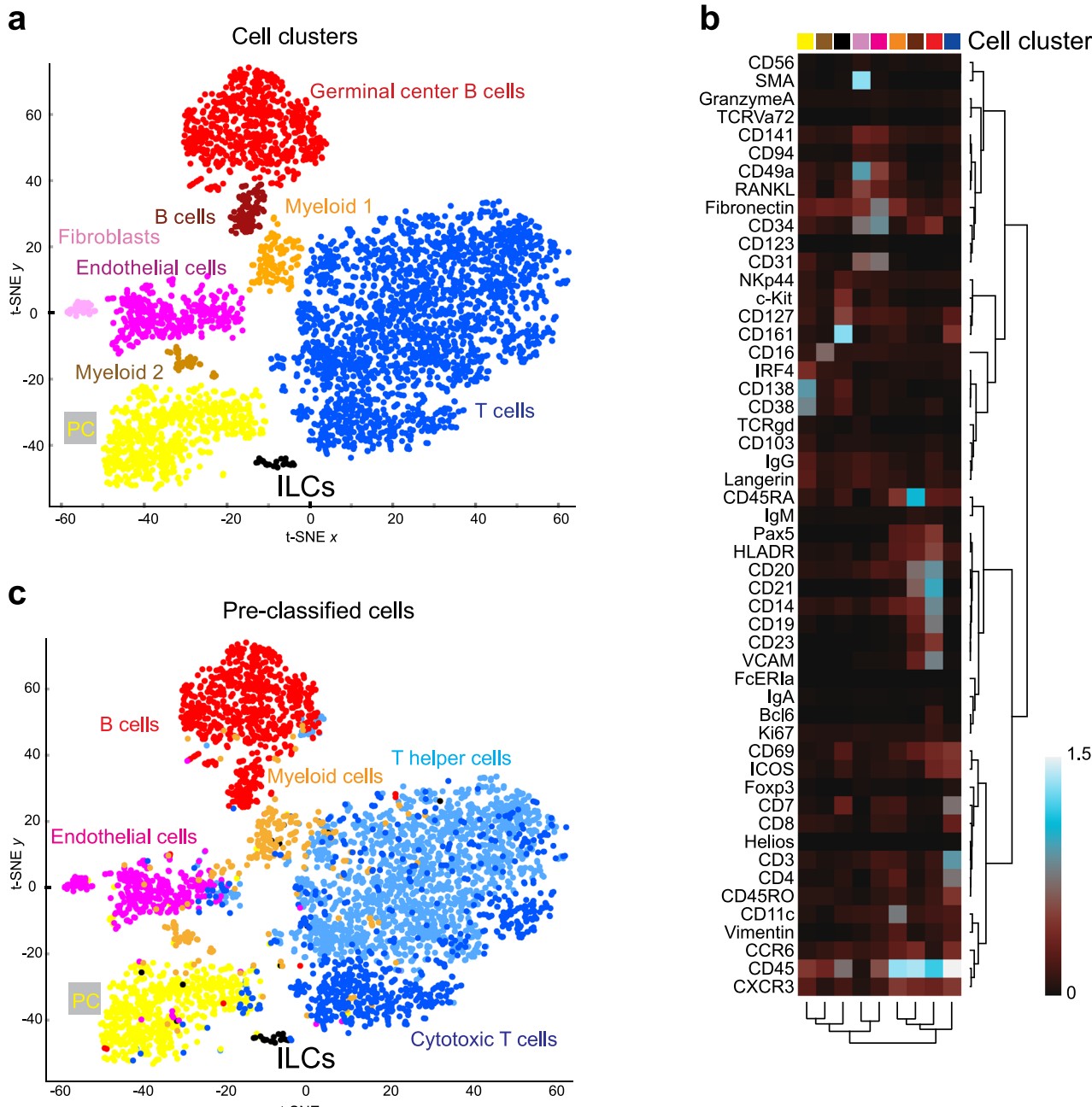

**Fig. 6 Clustering analysis of tonsil single-cell multiplexed histology data identifies ILCs and other relevant cell populations. a** The t-SNE map of tonsil single-cell MELC data reveals nine distinct clusters (germinal center B cells in red, B cells in dark brown, myeloid 1 in orange, myeloid 2 in light brown, T cells in blue, fibroblasts in pink, endothelial cells in magenta, plasma cells—PC—in yellow and ILCs in black) that are manually selected and annotated based on the expression profile shown in (**b**). Each dot represents one cell. **b** Heat-map representation of MELC data: mean relative expression level of the 52 markers analyzed for every cluster as in (**a**). **c** t-SNE map of tonsil single-cell MELC data previously classified into cell-types (B cells in red, T helper cells in light blue, cytotoxic T cells in dark blue, myeloid cells in orange, endothelial cells in magenta, PC in yellow, and ILCs in black), based on fluorescence thresholding of images for the lineage-defining markers. Each dot represents one cell. **a**–**c** (*n* = 5). See also Supplementary Fig.3.

compartment in the FOVs analyzed. These tissue compartments represent distinct functional areas exceeding the size of one FOV, that are consequently enriched in particular cell populations, making a pre-selection of tissue areas necessary for the histologic analysis of rare and not evenly distributed cell populations[30] such as ILCs. Notably, the deeper connective tissue septum that encapsulates each crypt is highly enriched in fibroblasts, ECM compounds like fibronectin and collagen, and it is characterized by lower cell density and high vascularization. This particular stromal composition is known to favor the survival and therefore,

the residency and function of tissue-resident populations such as plasma cells[31]. For this reason, we chose FOVs partly covering this area. While B cells and T helper cells represent the major immune cell types in our analysis, plasma cells and specially CD127+ ILCs represent rare populations, in line with previous reports[32,33]. Unlike plasma cells in close proximity or within the tonsillar germinal centers, which tend to be newly generated and short-lived, plasma cells within the connective tissue areas constitute a subset of higher maturity, higher Ig secretion, and a tendency towards residency and longevity[31,34,35]. Indeed, we

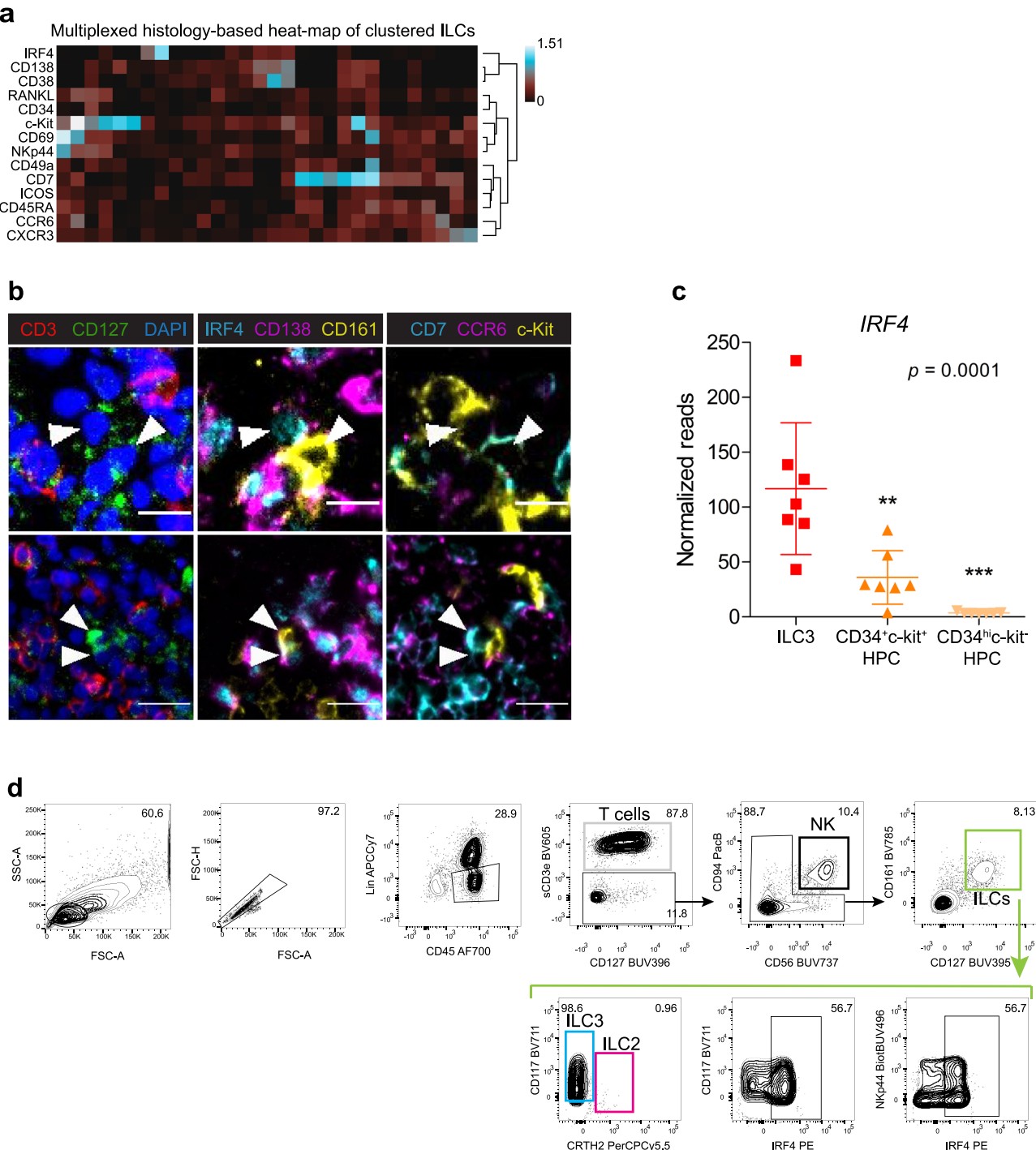

**Fig. 7 Clustering analysis of tonsil single-cell multiplexed histology data uncovers IRF4 expression in ILC3s. a** Heat-map representation of the relative expression level of 14 relevant markers analyzed for every clustered ILC identified in (Fig. 6a, b). **b** Overlay of MELC fluorescence images depicting relevant markers for the identification of ILCs (white arrows). Three representative examples of CD3−CD127+CD161+/−c-Kit+CCR6+/−CD7+/− ILC3s are shown in two different regions of interest of the tonsil (upper and lower panel). These ILC3 cells were allocated in the ILC cluster as shown in (**a**), and express IRF4 and/or CD138. CD3 (red), CD127 (green), and DAPI (blue) are shown in the first column. IRF4 (cyan), CD138 (magenta), and CD161 (yellow) are shown in the middle column. CD7 (cyan), CCR6 (magenta), and c-Kit (yellow) are shown in the right column. Scale bar: 10 μm in the upper panel and 20 μm in the lower panel. **a**, **b** (n = 5). **c** Dot plot depicting *IRF4* normalized transcript reads in sorted DAPI−Lin−CD94−CD127hiCD56+ ILC3s, CD34+c-Kit+ hematopoietic progenitor cells (HPC), and CD34hic-Kit−HPCs, extracted from the published microarray data[41]. Data are shown as mean ± SD and analyzed by one-way ANOVA ($F = 21.78$ and 2 *df*) with Bonferroni's multiple comparison test, where **$p < 0,005$ and ***$p < 0.0005$ (n = 7). **d** Representative complete gating strategy for flow cytometry analysis of tonsillar ILCs after magnetic depletion of CD3+ and CD19+ cells (n = 3) (see "Methods"). Lin− gate contains CD14−CD19−CD20−CD123−CD141−FcεR1α− cells· Lin−CD3−CD45+CD94+CD56+ are defined as NK cells (black). Lin−CD3−CD45+CD127+CD161+ are defined as helper ILCs (green), which are subclassified as c-Kit+CRTH2− ILC3s (98.6 %, cyan) or CRTH2+ ILC2s (0.96 %, magenta). IRF4 is shown to be co-expressed with the ILC3 markers c-Kit and NKp44 within the ILC population. See also Supplementary Fig. 4. Source data are provided as a Source Data file.

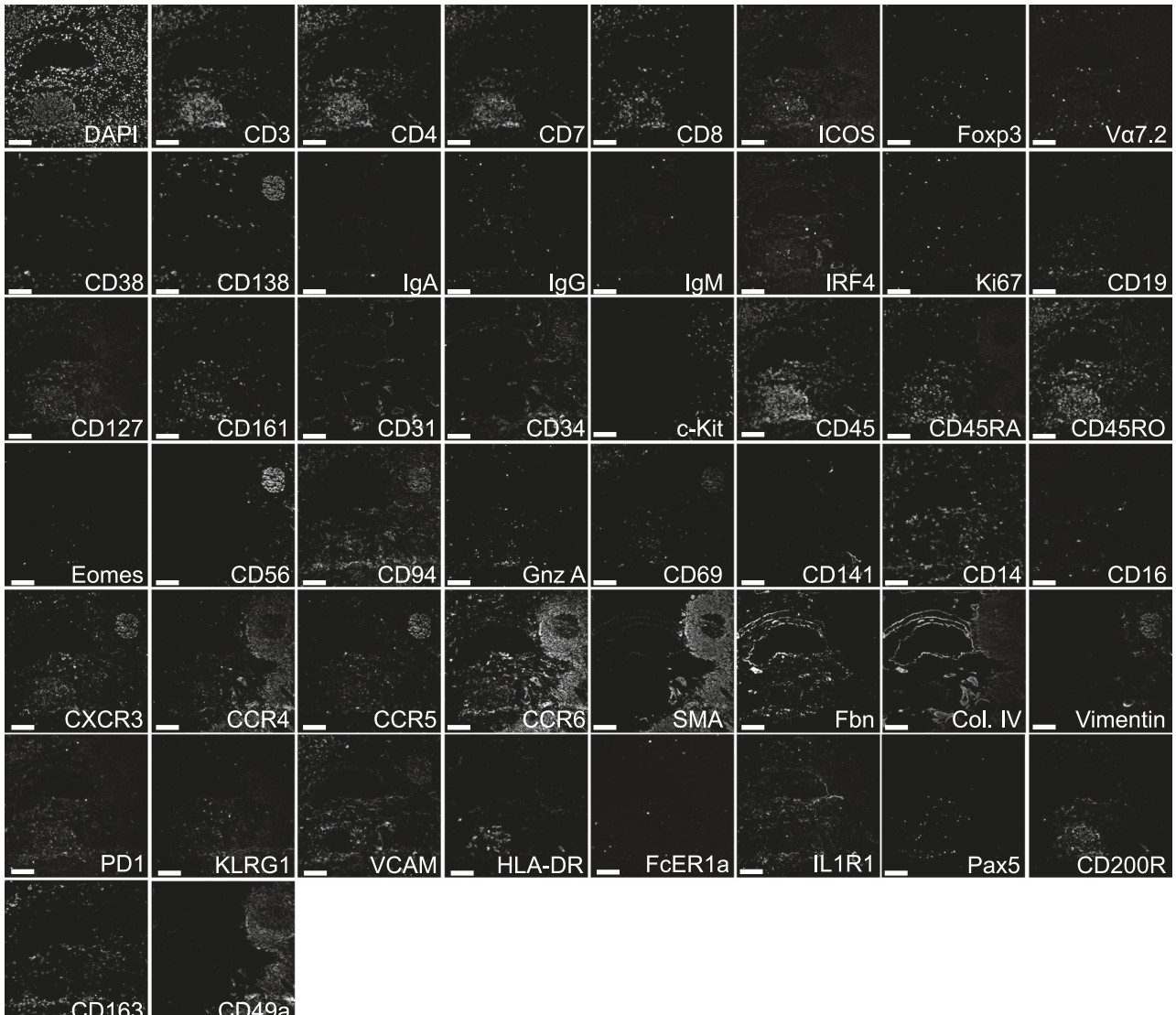

**Fig. 8 Panel overview of a 50 marker MELC run in a colon sample from a patient diagnosed with ulcerative colitis.** Each image shows the same field of view, sequentially stained with the depicted fluorescence-labeled antibodies, including surface proteins and transcription factors. Images contain 2048 × 2048 pixels and are generated using an inverted wide-field fluorescence microscope with a 20× objective, a lateral resolution of 325 nm, and an axial resolution above 5 μm. Gnz A granzyme A, Fbn fibronectin, Col. IV Collagen IV, SMA smooth muscle actin, Vα7.2 TCR Vα7.2. Scale bar: 100 μm.

observe plasma cells localized in the connective tissue to be non-proliferating cells and to contain high amounts of Ig, as evidenced by signal intensity. Importantly, we observe CD127[+] ILCs preferentially accumulating in the same tissue area and the micro-environment of ILCs in the tonsil is enriched in plasma cells, as well as in vessels and fibroblast-derived ECM proteins. A similar stromal niche composition has been described for ILC2s in several mouse tissues[13]. This preferential co-localization of both ILCs and plasma cells, together with higher expression of residency markers in this particular compartment, suggests that the connective tissue might provide niches for the residency and survival of various immune cell types, favoring CD127[+] ILCs to sit close to and potentially interact with plasma cells. Further functional studies are needed to support these observations, but in line with the work of several groups demonstrating the tissue-resident characteristics of ILCs[8] and the existence of ILC niches in several tissues[13,36,37], the histological approach used here also points towards the existence of microenvironments distinct in immune and stromal cell composition. ILCs in lymph nodes and spleen have been localized in the interfaces between T cell and B

cell regions, but not deep within B cell follicles or T cell zones, and, there, they might interact with both B and T cells thereby shaping adaptive immunity[37–40]. Similarly, we find ILCs in human tonsils to localize around B cell follicles, but not within, in close contact to the fibronectin ring that lines and shapes such structures.

The main challenge for the identification of ILCs in situ is the need for extensive antibody panels. Although we overcame the limitation of standard immunofluorescence techniques by the multiparametric nature of our approach, we were still facing to some extent subjective image thresholding steps based on visual inspection, and complex and long CP pipelines for the classification of cell types of interest. Importantly, we show that less biased and time-demanding analysis approaches can also be successfully coupled to MELC. We identify here several tonsillar cell populations by clustering analysis using all data dimensions simultaneously. Cell clusters are consistent with data where cell classifications are performed beforehand, based on fluorescence thresholding. The spatial distribution of the clustered cells and the pre-classified cells precisely depicts the original

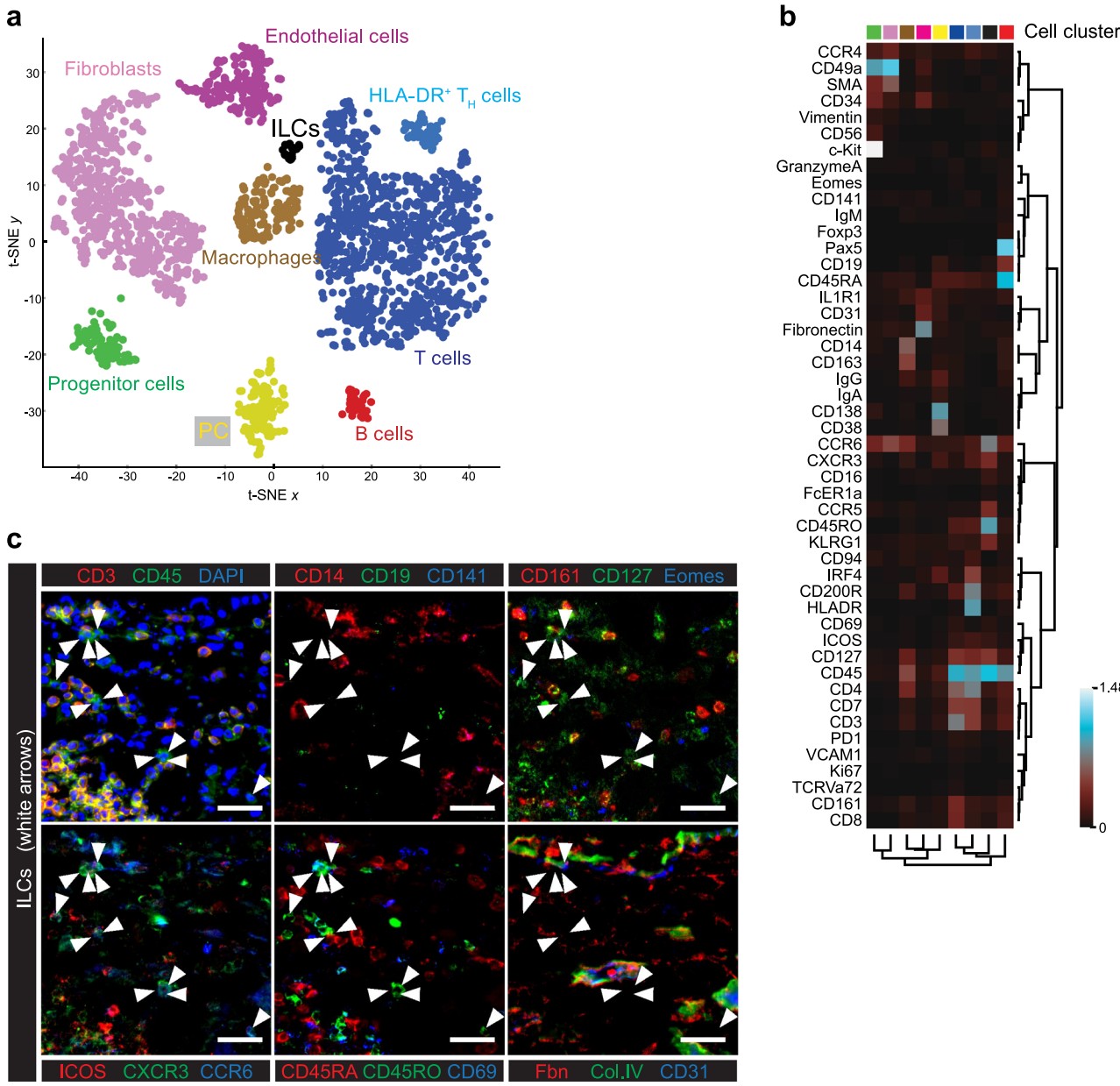

**Fig. 9 Clustering analysis of colon single-cell multiplexed histology data reveals a distinct ILC population. a** t-SNE map of all segmented cells expressing at least one of the markers included in the panel (cut-off = 0.1 MFI). Nine phenotypically distinct clusters are manually selected and annotated based on their expression profile as in (**c**) and depicted in a color-coded fashion (fibroblasts in pink, endothelial cells in magenta, progenitor cells in green, plasma cells in yellow, B cells in red, myeloid cells in brown, HLA-DR+ T cells in cyan, T cells in blue and ILCs in black). Each dot represents one cell. **b** Heat-map representation of the mean relative expression levels for the 48 markers analyzed in every cluster as in (**a**). **c** Region of interest (ROI) shown as an overlay of fluorescence images of relevant markers for ILC characterization. White arrows indicate ILCs. CD3 (red), CD127 (green), and DAPI (blue) are shown in the upper left panel. CD14 (red), CD19 (green), and CD141 (blue) are shown in the upper middle panel. CD161 (red), CD127 (green) and CD141 (blue) are shown in the upper right. ICOS (red), CXCR3 (green), and CCR6 (blue) are shown in the bottom left panel. CD45RA (red), CD45RO (green), and CD69 (blue) are shown in the bottom middle panel. Fibronectin (red), Collagen IV (green), and CD31 (blue) are shown in the bottom left panel. Scale bar 20 μm. **a**–**c** (n = 2).

immunofluorescence images, supporting both cell classification strategies. Remarkably, pre-classified CD127+ ILCs mainly confirm a distinct cluster in the t-SNE map, proving clustering analysis of multidimensional image data effective for the in situ identification of rare cell subsets with challenging cell profiles. We found ILCs in tonsils to broadly express c-Kit and, to a lesser extent, CD7 and NKp44, indicating an ILC3-enriched population in this tissue, in line with previous studies[6]. Moreover, we detected the expression of the TF IRF4 in the nuclei of a fraction of ILC3s, in all tonsil data sets analyzed by MELC. Confirming

our findings on a transcriptional level, we found ILC3s to express *IRF4* transcripts at significantly higher levels than their lineage-specified CD34+c-Kit+ hematopoietic progenitors (HPCs) and the differences were even higher when compared to the CD34hi c-Kit- HPCs, known to be a less restricted ILC progenitor population[41]. This might suggest that the acquisition of this marker is linked to the maturation and/or homeostasis of ILC3s, at least in the tonsil, similar to IRF4 functions in B and T cells. Importantly, a previous publication based on scRNA-seq reported a strong enrichment in *IRF4* gene expression in tonsillar ILC3s

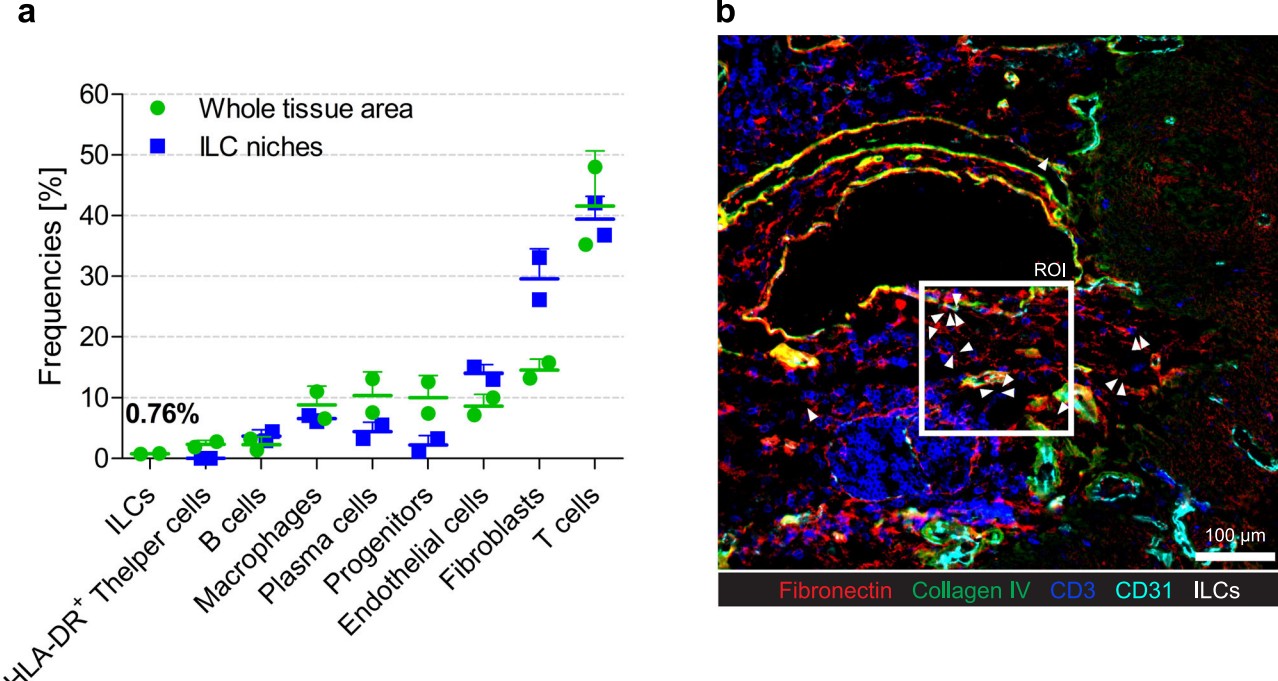

**Fig. 10 ILCs show a particular and conserved localization pattern with respect to stromal landmarks. a** Bar graph shows the distribution of several hematopoietic and structural cells as identified in Fig. 9, in the whole tissue area analyzed compared to their distribution within ILC niches. Data are shown as mean ± S.D. **b** Overlay of fluorescence images emphasizing ILC distribution throughout the tissue sample, showing Fibronectin (red), Collagen IV (green), CD3 (blue), and CD31 (cyan). ILCs are depicted as white arrows. The ROI depicts the area and the localization of several ILCs as shown in (Fig. 9c). **a**, **b** (n = 2). Source data are provided as a Source Data file.

compared to tonsillar ILC1s, ILC2s, and NK cells[3], further supporting IRF4 as a marker for tonsillar ILC3s. At the protein level, we also show IRF4 expression on tonsillar ILC3s by flow cytometry. Since this TF plays essential roles in both plasma cell and T cell effector functions[42], it is compelling to study its function in ILCs in the future.

Our histology data also revealed the expression of the plasma cell-related marker CD138 on tonsillar ILCs. We are aware that the limited resolution and the need for cell segmentation inherent to our multiplexed histology approach might lead to signal cross-contamination of adjacent and tiled cells, even in high-resolution images. This is particularly relevant in this case since we find CD127+ ILCs and plasma cells to localize close to each other, in distinct microanatomical areas of the tonsils. Although we also detected *SDC1* transcripts in sorted ILC3 at significantly higher levels than in the CD34hic-Kit- HPC population, thus supporting our histology data, we were not able to confirm CD138 protein expression on ILCs by flow cytometry. CD138 is a cell surface proteoglycan susceptible to shedding during the preparation of single-cell suspensions from tonsils, and discrepancies between flow cytometry and immunofluorescence with regard to detection of CD138 in plasma cells, which express high levels of this proteoglycan, have been previously reported by others[43,44]. Taken together, further studies are needed to verify our findings regarding the expression of CD138 in ILCs and understand the discrepancies observed between the histology/transcriptional data and the flow cytometry data.

The fact that ILCs and plasma cells in the tonsils, which are localized in close proximity and accumulate in the same micro-anatomical areas, share expression patterns, points to high tissue compartmentalization not only in terms of cellular distribution but also in terms of molecular fingerprints. This highlights the importance of protein topology within tissues in relation to function. However, molecular commonalities within tissue compartments are overseen in other experimental settings, where the spatial resolution cannot be preserved and might be important to gain a deeper understanding of how tissue micro-environments shape cell phenotypes and vice versa. This is particularly relevant for cell types such as ILCs, known to be highly heterogenic and plastic, a feature that allows them to adapt to changing environmental cues.

We also identify CD127+ ILCs integrating a distinct cluster in ulcerative colitis intestines. In contrast to our observations in the tonsil, clustered CD127+ ILCs did not express IRF4 or CD138. Interestingly, both chronically inflamed mucosal tissues share distinct molecular patterns that help us predict ILC localization. We do observe single ILCs in all tissues/samples analyzed. However, and despite their rare abundance, the majority of CD127+ ILCs tend to localize in groups and undergo intimate contact with conduit ducts and ring-like structures, characterized by collagen IV and fibronectin fibers that coat the vessels. Taken together, this suggests the existence of particular microenvironments, which promote their preferred accumulation. It is important to note that our analysis is currently performed in 2D and, therefore, the groups of CD127+ ILCs may be more numerous in the natural 3D tissue structure. In addition, their particular microanatomical location would support two main roles for CD127+ ILCs in tissues. On the one hand, this position would enable them to monitor afferent lymph and/or blood for cytokine signals, in order to rapidly mount an appropriate response within the tissue and, on the other hand, to exert a systemic effect by interacting with efferent vessels in case that tissue integrity is in jeopardy. It remains to be clarified whether CD127+ ILCs are preferentially located close to blood or to lymphatic vessels and to further functionally characterize ILC-stromal interactions that potentially occur. In light of the findings shown here, both questions will be addressed in future studies.

In conclusion, we show here a customized analysis pipeline based on highly multiplexed immunofluorescence data, suitable for the identification and characterization of ILCs and their microenvironments. Thereby, we highlight IRF4 as a marker expressed by tonsillar ILC3s. With our spatially resolved approach, we show that tonsillar CD127$^+$ ILCs and plasma cells are localized in the same tissue areas and pinpoint stromal landmarks for CD127$^+$ ILC localization. Thus, we define characteristic microenvironments for ILCs, which are conserved across several tissues, suggesting the existence of defined tissue patterns constituting ILC niches. The work presented here establishes the basis for future histological analysis of such rare immune cell populations, allowing a better understanding of the role of ILCs in both tissue homeostasis and inflammation.

## Methods

**Human mucosal tissues.** Tonsils from patients undergoing tonsillectomy and colon samples from patients diagnosed with ulcerative colitis were received fresh after written informed consent, according to the Declaration of Helsinki. Approval by the medical ethics commission of the University of Freiburg (251/13_140389) and the ethics commission of the Charité-Universitätsmedizin Berlin (EA2/078/16) were obtained, in accordance with the local ethical guidelines.

**Tissue preparation for MELC.** Fresh frozen tissue was cut 5 μm thick with a NX80 cryotome (ThermoFisher, Waltham, Massachusetts, USA) on 3-aminopropyltriethoxysilane (APES)-coated cover slides (24 × 60 mm; Menzel-Gläser, Braunschweig, Germany). Samples were fixed for 10 min at room temperature using a freshly opened EM grade PFA ampulla (methanol- and RNAse-free; Electron Microscopy Sciences, Hatfield, Philadelphia, USA) diluted to 2%. After washing three times with PBS, samples were permeabilized with 0.2% Triton X-100 in PBS for 10 min at room temperature. Subsequently, a blocking step with 10% goat serum and 1% BSA in PBS was performed for at least 20 min. Afterwards, a fluid chamber holding 100 μl of PBS was created using "press-to-seal" silicone sheets (Life technologies, Carlsbad, California, USA; 1.0 mm thickness) with a circular cut-out (10 mm diameter), which was attached to the coverslip, surrounding the sample.

For every MELC run, a bottle of fresh PBS with 1% BSA and 0.02% Triton X-100 was used. The sample was placed on the sample holder and fixed with adhesive tape followed by accurate positioning of the binning lens, the light path, as well as Köhler illumination of the microscope.

**MELC image acquisition.** We generated the multiplexed histology data on a modified Toponome Image Cycler® MM3 (TIC) originally produced by MelTec GmbH & Co.KG Magdeburg, Germany[16]. The robotic microscopic system consists of: (i) an inverted widefield (epi)fluorescence microscope Leica DM IRE2 (20 × /0.8 NA objective air lens, filter setup: Omega Optical XF116$^-$2, AHF F46-010, AHF F46-009, and AHF F46-000) equipped with a CMOS camera (Orca®-flash4.0 LT, Hamamatsu Photonics GmbH, 2048 × 2048 pixels, pixel size 6.5 μm, no binning) and a motor-controlled XY-stage, (ii) CAVRO XL3000 Pipette/Diluter (Tecan GmbH, Crailsheim, Germany), and (iii) a software MelTec TIC-Control for controlling microscope and pipetting system and for synchronized image acquisition.

The MELC run is a sequence of cycles, each containing the following four steps: (1) pipetting of the fluorescence-coupled antibody onto the sample, incubation, and subsequent washing; (2) cross-correlation based auto-focusing, which compares the current phase-contrast images with a phase-contrast reference image acquired at the beginning of each MELC run, defines the xyz position of the FOV of interest within the whole sample, and thus corrects displacements in xyz for the aligned acquisition of 3D fluorescence image stacks for each marker (±7 z-steps; z-step = 1 μm); (3) photo-bleaching of the fluorophore using the optimal time span to minimize the fluorescence signal of each staining followed by washing of the specimen; and (4) a second autofocusing step followed by the acquisition of a 3D stack post-bleaching fluorescence image.

In each four-step cycle, up to three fluorescence-labeled antibodies were used, combining PE, FITC, and DAPI. After the sample was labeled by all antibodies of interest as described above, the experiment is completed.

**MELC antibody panel.** The antibodies used for multiplexed immunofluorescence histology of tonsil and colon samples are listed in Supplementary Table 2. The antibodies were stained in the indicated order. Steric hindrance issues that might appear due to this specific labeling order have been ruled out as previously shown[45].

**Image pre-processing.** Since the technology is based on sequential staining of one single section and due to the mechanic tolerance of the motorized microscope table, each data set had to be registered by cross-correlation. For that, all images

were aligned based on the reference phase-contrast image taken at the beginning of the measurement (Supplementary Fig. 1a). Afterwards, we processed each fluorescence MELC image by background subtraction and illumination correction (Supplementary Fig. 1b)[16]. Background subtraction was accomplished by subtracting the bleaching image of one cycle from the fluorescence image of the following cycle, where both bleaching and fluorescence images were acquired under the same conditions. Thereby, tissue auto-fluorescence and the potential residual signal from the previous cycle were removed. In the case of uncoupled antibodies and to account for the unspecific signal of the secondary antibody used, we subtracted the fluorescence image of the secondary antibody stained and acquired before the corresponding primary antibody, instead of the bleaching image. Illumination correction of each fluorescence image was performed, based on the bleaching image of the same cycle, where a cubic spline interpolation over the minimum intensity values was used to approximate the function accounting for uneven illumination and/or detection. In order to account for slice thickness and for the fact that the slice is not perfectly aligned with the focal plane of the objective lens, e.g., due to ripples in the tissue, an "extended depth of field" algorithm was applied on the 3D fluorescence stack in each cycle. In this way, the signal originating from the focal plane of each image of the 3D fluorescence stack was projected to a single plane resulting in a single 2D fluorescence image[46]. Images processed in that way were publicly available in the Zenodo open access repository https://zenodo.org/ (see "Data availability").

Images were then normalized in ImageJ[47], where a rolling ball algorithm was used for background estimation[48], edges were removed (accounting for the maximum allowed shift during the autofocus procedure) and fluorescence intensities were stretched to the full intensity range (16 bit = >2$^{16}$) (Supplementary Fig. 1c). The 2D fluorescence images generated in this way were subsequently segmented and analyzed.

**Cell segmentation and single-cell feature extraction.** Segmentation was performed in a two-step process, a signal-classification step using Ilastik 1.3.2[18] followed by an object-recognition step using CellProfiler 3.1.8[19], as described elsewhere[27]. Ilastik was used to classify pixels into three classes (nuclei, membrane, and ECM) and to generate probability maps. A combination of images was summed up and used to classify membranes and ECM, while only the DAPI image was used to classify nuclei. The random forest algorithm (machine-learning, Ilastik) was trained by manual pixel-classification in a small region of a data set (~6% of the image). The rest of the data set, as well as four extra data sets analyzed here, were classified without re-training, using the random forest algorithm. In this way, three types of images were generated: a nuclei probability map, a membrane probability map, and an ECM probability map, the latter containing all pixels that do not belong either to the nuclei or to the membrane probability maps. CellProfiler was subsequently used to segment the nuclei and membrane probability maps and to generate nuclei and cellular binary masks, respectively. These masks were superimposed on the individual fluorescence images acquired for each marker included in the MELC run, in order to extract single-cell information for individual markers, i.e., mean fluorescent intensity (MFI) of each marker per segmented cell. Complete and detailed CellProfiler pipelines and all data tables generated are publicly available in the Zenodo open access repository https://zenodo.org/ (see "Data availability").

**Cell classification and neighborhood analysis based on fluorescence thresholding of images.** Based on the extracted single-cell MFI values, cells were classified into known cell types based on expected marker combinations and their occurrence in different tissue compartments was quantified. ILC niches were defined as areas of 10 μm radius around each ILC, based on the assumption that this equals the average diameter of immune cells. Taking ILCs as seed-points, pixels were expanded until covering these neighborhood areas and the niches were segmented as objects. Pre-classified immune cell types of interest and pre-segmented ECM structures were quantified within the ILC niches. Complete and detailed CellProfiler pipelines are publicly available in the Zenodo open access repository https://zenodo.org/ (see "Data availability").

**Cell classification based on clustering algorithms.** Segmented cells were analyzed in Orange 3.26.0[49] using several algorithms for dimensionality reduction, which use the list of MFI values as single-cell features, irrespective of the nature or function of the investigated markers. Cluster analysis of the tonsil single-cell data was performed by analyzing segmented B cells (CD45$^+$CD19$^+$), plasma cells (CD138$^+$), T helper cells (CD45$^+$CD3$^+$CD4$^+$), cytotoxic T cells (CD45$^+$CD3$^+$CD8$^+$), cells of myeloid origin (CD11c$^+$ and/or CD14$^+$ and/or CD56$^-$CD16$^+$ and/or CD141$^+$), endothelial cells (CD31$^+$) and ILCs (CD45$^+$CD127$^+$CD3$^-$CD19$^-$CD14$^-$CD123$^-$CD141$^-$FcεRIα$^-$). Clustering of the colon single-cell data was performed by analyzing all segmented cells expressing at least one marker by setting a cut-off at 0.1 MFI. T-stochastic neighbor embedding (t-SNE) was performed with 52 principal components for the tonsil data sets and 48 principal components for the colon data sets, a perplexity of 30 and 1000 iterations.

**Data transformation.** Data were transformed using the hyperbolic arcsine function with a scale argument of 0.2 previous to cluster analysis.

**Data normalization**. Fluorescence intensities per pixel were normalized to the full 16-bit range in ImageJ and brought to a 0–1 scale in CellProfiler.

**Statistical analysis**. Statistical analysis was performed with GraphPad Prism® 5.04 (Graph Pad Software, LA Jolla, CA, USA). Significance was assessed by one-way or two-way ANOVA followed by Bonferroni multiple comparison test. Exact $p$ values are depicted within each graph when possible or, alternatively, ranges are described in each figure legend.

**Cell isolation**. Tonsils were thoroughly dissociated in sterile PBS/BSA/2 mM EDTA and passed through a 70 μm strainer. Mononuclear cell suspensions were isolated by Ficoll density gradient centrifugation and MACS-depletion of $CD19^+$ and $CD3^+$ cells (Miltenyi Biotec) was performed prior to staining.

**Flow cytometry**. A single-cell suspension of tonsil mononuclear cells was stained for surface markers in 100 μl PBS/BSA for 15 min at 37 °C. After washing, biotin was stained with fluorochrome-labeled streptavidin for 10 min at 4 °C. Afterward, cells were fixed for 35 min at RT, permeabilized, and TF were stained at RT for 45 min using the Foxp3/TF staining buffer set (eBioscience). Measurements were performed on a FACSymphony (BD Bioscience) and data analyzed with FlowJo v.10.6.01 software (TreeStar). The antibody panel used for flow cytometry is shown in Supplementary Table 3. Flow cytometry was performed according to *Guidelines for the use of flow cytometry and cell sorting in immunological studies*[50].

**Microarray analysis**. Gene expression analysis was performed on tonsillar ILC3s and progenitor populations from data generated in[41] and as described therein. See the data availability section.

**Reporting summary**. Further information on research design is available in the Nature Research Reporting Summary linked to this article.

## Data availability

All source data (source images) and materials (Cell Profiler pipelines) associated with this study are publicly available on the Zenodo open−access repository https://zenodo.org/: the tonsil data set under https://doi.org/10.5281/zenodo.3744152, the colon data set under https://doi.org/10.5281/zenodo.3744173 and Cell Profiler pipelines and complete single-cell data tables under DOIs https://doi.org/10.5281/zenodo.3744206 and https://doi.org/10.5281/zenodo.3744273, respectively. All data sets (all DOIs) are interlinked within the public repository and, therefore, easy to access.

The raw data, as well as log2 transformed normalized values for the microarray analysis, are available under the GEO accession number GSE63197. Source data are provided with this paper.

## Code availability

Custom codes were not developed for this work and the references for the software used are provided in the methods.

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

## Acknowledgements

We would like to thank Daniel Wendisch (Immunology of Infectious Diseases & Vaccinology group, Charité—Universitätsmedizin, Berlin) for helpful discussions. This study was supported by funding from the Deutsche Forschungsgemeinschaft, (SPP1937, HA5354/8-2 to A.E.H.), TRR 130 C01 and SFB 1444 P14, to A.E.H. and R.N. The contribution of Max Seidl was funded in part through CRC 1160 traveling grants. L.P. was supported by CRC854, Z01. The work was supported by an "Adding 3R Value" grant of Charité 3R to A.E.H. The authors affiliated with Charité for the duration of this study acknowledge that Charité is a corporate member of Freie Universität Berlin and Humboldt-Universität zu Berlin.

## Author contributions

A.E.H. and A.P.R. conceptualized the study. K.K., M.S., C.R., and D.C.H. provided samples. R.M., S.B., R.U., D.C.H., K.H., L.P., and A.P.R. performed experiments. R.M., R.K., and A.P.R. analyzed the data. A.E.H. and A.P.R. interpreted the results and wrote the manuscript. R.K., S.B., K.K., W.M., M.S., C.R., and R.N. discussed results and reviewed the manuscript.

## Funding

## Competing interests

W.M. is an employee (Manager of Bioinformatics) of Miltenyi Biotec B.V. & Co. KG, Bergisch Gladbach, Germany. Lars Philipsen is shareholder of M04 Patent—und Beteiligungsgesellschaft Ltd., Magdeburg, Germany. The remaining authors declare no competing interests.
