## [Peer Review File · Nature Communications]

REVIEWER COMMENTS

Reviewer #1 (ILC, transcription factor regulation) (Remarks to the Author):

In this report, Pascual-Reguant and colleagues use highly multiplexed fluorescence histology, based on serial staining and bleaching of histology slides, to characterize innate lymphoid cells (ILCs) in samples of human tonsils and colon tissue. Elaborate image analysis is then developed to determine the cell types visualized by the combinations of 53 markers, define structures in tonsils and colon samples, and identify ILC niches through characterization of the immediate neighborhood. Finally, the combination of the 53 markers and the ensuing image analysis allows the authors to define clusters, i.e. cell types, and finely determine their localization in the tissue.

This is primarily a technical paper, which combines big data style of analysis with complex image analysis, and is therefore very interesting for the field. For this part, however, I am unable to assess the accuracy of the approach.

In terms of immunology, this study does not add much to the current knowledge, except that ILCs appear clustered close to plasma cells in tonsils, and that a subset of ILCs appear to express markers of plasma cells (the C4a cluster in Fig. 5). This latter claim is however weakly substantiated. The few markers used to define cluster C4a are not sufficient to claim that these cells are ILCs. Thus, the authors should do much more to verify this claim, for example by flow cytometry.

Reviewer #2 (Imaging immune systems, signaling) (Remarks to the Author):

The authors present a multiplexed fluorescence technology based on cycles of staining and bleaching with mostly PE labeled antibodies, imaging, image registration, segmentation, measurement and analysis of single cells. The target of the analysis is "helper" ILC, a rare population overall, and an important subset to localize in human tissues requiring multiplexing. As a rare population, ILC are expected to be surrounded by other cells. The pipeline seems robust and is open source so potentially accessible to a wider range of scientists to adopt on their own without committing to an expensive commercial platform. I would have appreciated a more explicit discussion of how the analysis pipeline copes with the obvious problem that due to limitations of resolution and membrane to membrane tiling of cells that cells are contaminated with signals from adjacent cells. I have some concerns the authors can address.

There are two aspects of the results that give me some concerns about an influence of possible cross-contamination on the system. First, they comment that ILC are clustered. I would expect this if there was a bias against detection of single ILC surrounded by lineage positive cells. Second, the a couple of cases are raised where ILC in different locations share markers characteristic of their microenvironment. This was particularly surprising for the plasma cell marker CD138. The authors defend this result by saying that other B cell markers like Ig were not seen in the same ILC that are near plasma cells, but I think some independent validation that CD138 on ILC is needed- which could be done by multiparameter flow cytometry experiment or scRNAseq. My feeling is that the authors, and anyone trying to generate single cell datasets from immunofluorescence histology, needs to recognize this problem, quantify it as best they can, and then take advantage of it rather they trying to bury it. If the authors accept that the CD138 may be cross-contamination from neighboring plasma cells that is not getting filtered out and even verify that this is true, then they can use the contamination as an independent way to assess possible cell-cell contacts in the dataset. Looking at the data details presented it also looks like some membrane proteins have strong intracellular signals,

maybe due to biosynthesis, turnover or storage, but these will be easier to assign to cells than proteins that are mostly at the plasma membrane. The IRF4 signal in the nucleus of some cells with associated CD127 signals is also clear. But validation of the unexpected combinations with a different method would build confidence in the novel conclusions.

One validation idea that might address a number of issues- take a single cell suspension of tonsil cells, attach them to PLL coated glass as low density (all cells well separated) and high density (all cells overlapping) and then run the pipeline. Does the analysis come out the same for analysis of the same number of cells well separated vs crowded at tissue density. I think this would be a good single cell ground truth and assessment of cross contamination due to shared cells borders. Perhaps this is better than flow cytometry as it allows use of same reagents and really tests the pipeline.

Reviewer #3 (Computational analyses, systems biology) (Remarks to the Author):

The authors present a study using Multi Epitope Ligand Cartography (MELC) images of the human tonsil. This highly multiplexed imaging technique allows them to identify a number of cell types and to localize them in the tissue. The authors then focus on the analysis of the spatial distribution of Innate Lymphoid Cells (ILCs), which they address with two computational approaches with varying degrees of supervision.

Understanding tissue composition by the analysis of cell types in their spatial context is an important challenge today, for which a variety of imaging and analysis techniques exist, namely the many variants of multiplexed FISH or imaging mass cytometry. MELC is another technique to overcome the limitation in the number of simultaneously imaged markers. Proposed in 2006, MELC might nicely complement the arsenal of tools in this regard. The application of MELC to analyze ILCs is certainly a novel application, even though the multiplexed analysis of tonsil sections (imaged with a different technique) has been published before.

In principle, I found the article interesting and timely. At the methodological level (which is the expertise I can contribute to the review process), I did not find much novelty: the authors use standard tools to perform image processing and analysis (background correction, pixel classification, object segmentation from the probability maps). A very similar version of the overall workflow has actually been proposed previously, even for the analysis of multiplexed images of tonsil sections, albeit for a different imaging technique [1]. On the other hand, the proposed method is solid (with some caveats mentioned below) and clearly serves the biological conclusions.

The article in its present form has the following (potential) weaknesses:

- It is not very clear to me what the major contribution of the article is (biology, imaging or analysis). If the main result is on the imaging side, a deeper discussion of the advantages and drawbacks of MELC for this kind of application would be very helpful for the community. Regarding the analysis part: the computational aspects are handled correctly (with the exceptions mentioned below), but are not by themselves sufficiently new to justify publication. The novelty with respect to [1] seems limited to me.
- The distinction between "hypothesis-driven" and "data-driven" in this context seems misleading to me. In the first approach, cell types are identified by rules formulated from the literature (i.e. expression of cell type specific marker genes). In the second approach, clustering is used to first identify all cell types, and then to characterize them according to rules from the literature (by inspection of the heatmap). In both approaches, prior knowledge is thus used to make sense of the

data, in one case in an entirely supervised way, in the second case by cluster interpretation. Concretely, what would be the hypothesis used in the first approach, which is not used in the second? Also, I do not really understand what the second approach brings in addition to the first. If all cell types are known, it seems to me that the problem is effectively supervised and not unsupervised. If there was a new cell type or if the clustering was really essential to draw biological conclusions, this would be much more convincing. I elaborate on this below.

- The analysis of the ILC niches is in principle the most interesting part of the article, but here the authors should stratify the niches according to the region where they are situated instead of comparing them to an overall statistic (see comment below).

Other than that, I think the article illustrates very nicely what can be gained from spatial analysis. Also, the image data is of very high quality, and this sets this work apart from some of the previous studies, where only low-resolution images could be obtained.

Thomas Walter.

[1] (1) Durand, M.; Walter, T.; Pirnay, T.; Naessens, T.; Gueguen, P.; Goudot, C.; Lameiras, S.; Chang, Q.; Talaei, N.; Ornatsky, O.; Vassilevskaia, T.; Baulande, S.; Amigorena, S.; Segura, E. Human Lymphoid Organ CDC2 and Macrophages Play Complementary Roles in T Follicular Helper Responses. *The Journal of Experimental Medicine* 2019, 216 (7), 1561 LP – 1581. <https://doi.org/10.1084/jem.20181994>.

Detailed comments:

1. Normalization procedure: the authors propose to stretch the dynamic range of each individual image (page 5). This means that low intensity channels (for instance due to low expression) will be multiplied by a much larger factor than high intensity channels, which in turn can have substantial impact on the clustering procedure. The authors should motivate their choice, explain potential caveats and also provide some data to get a feeling for the impact on the clustering result.
2. Otsu-threshold: Random Forests provide posterior probabilities as output, i.e. the logical threshold would be 0.5 for a binary segmentation problem, or a simple max for a multi-class problem. It is not clear to me, why the authors would apply an Otsu-threshold on the posterior probability map.
3. The procedure of assessing whether a cell is positive with respect to a particular marker was not entirely clear to me. The authors write "Subsequently, negative and positive cell populations for each marker were identified by intensity thresholding in order to classify cell-types, similar to conventional gating strategies performed in flow cytometry analysis." How did the authors set these thresholds? (e.g. by visual inspection of the images, or visual inspection of the histograms, etc.)
4. "Thus, ILCs were automatically classified and annotated as Lin- CD45+CD127+ cells.". First, I do not understand what the authors mean by "classify and annotate" (several occurrences in the text). Do they mean that the rule set Lin- CD45+CD127+ defines candidates that are then manually annotated by an expert? Or do they simply mean that ILCs were defined as Lin- CD45+CD127+ cells (which corresponds to applying thresholds in the respective channels)? If manual annotation is involved, what additional information would be used by the human annotator?
5. Analysis of ILC niches: in principle, this is the most interesting part of the study, because the spatial configurations can only be analyzed with image data (unlike cell type clustering and cell type calling, which can be done with molecular techniques). However, there is an issue with the statistical test. The authors compare the occurrences of cell types in the niches with occurrences outside. The problem with this is that the overall spatial distribution of cell types is very heterogeneous and compartmentalized. If we want to test whether the niches have a particular composition, one would have to compare to the distribution in a similar region (e.g. B-cell-region, crypt, T-cell-region, etc.).
6. "This tonsil region constitutes the tissue compartment that showed greatest statistically significant

differences in terms of ILC accumulation, with p values ranging from 0.05 to 0.005, compared to the other tissue areas analyzed.” – The sentence is not very clear to me. Do the authors want to say that p-values of ILC accumulation vary most in the subepithelial connective tissue? What is the biological conclusion from this result?

7. Clustering approach:

a. Like all clustering approaches, the final results are heavily impacted by the normalization procedure and correlation structure of the data. Often, we have to accept these shortcomings, because we are in a truly unsupervised setting, but this is not the case here: the authors perfectly know which combination of expression values defines the cell types. The limitation of these unsupervised approaches should at least be mentioned. For instance, I think that the number of 4 clusters is actually one among several possible solutions and might just represent the correlation structure of the chosen markers.

b. The authors analyze the cluster distribution of ILCs. I would find it much more convincing to isolate all ILCs (supervised approach) and to provide hierarchical clustering using all markers for this subset of cells. This would have been a much more direct approach.

c. In principle, there is nothing wrong with the analysis itself, but I somehow feel that it is not presented in the right light. The method is not truly unsupervised, as the markers are chosen and the clusters are interpreted according to prior knowledge; it is merely a way of grouping the data so that we can better impose our prior knowledge without extensive cell annotation. For instance, the authors state that “the data-driven approach presented here is suited for the identification and characterization of rare ILCs in several human tissues”, but this could have been achieved with their supervised approach.

Minor comments (typos, expression) :

- Page 6 : “Stroma-derived ECM proteins, such as fibronectin and smooth muscle actin (SMA) were also segmented and identified.”: the sentence is misleading (not the proteins are segmented).

- Page 6: “an area of ... was segmented”: technically, we would not speak of segmentation here, as the 10 μ m neighbor region was determined without taking the image signal into account. I suggest to replace “segmented” by “defined”.

- “... and pixels on the other four data-sets were classified in an unsupervised manner” Actually, this is a misnomer. The segmentation method is supervised, and application of a trained classifier on new data does not make it unsupervised. The authors mean “without retraining”.

- “... and minimizes user-based variability in the analysis.” - I am not sure about this statement.

Clearly, all automatic methods remove user-bias to some extent, and nobody would analyze these data completely manually. Among all automatic segmentation methods however, pixel-classification methods are not less user-biased (at the contrary). I would suggest to remove this half-sentence.

Point-to-point response to reviewers' comments

Reviewer #1 (ILC, transcription factor regulation) (Remarks to the Author):

In this report, Pascual-Reguant and colleagues use highly multiplexed fluorescence histology, based on serial staining and bleaching of histology slides, to characterize innate lymphoid cells (ILCs) in samples of human tonsils and colon tissue. Elaborate image analysis is then developed to determine the cell types visualized by the combinations of 53 markers, define structures in tonsils and colon samples, and identify ILC niches through characterization of the immediate neighborhood. Finally, the combination of the 53 markers and the ensuing image analysis allows the authors to define clusters, i.e. cell types, and finely determine their localization in the tissue. This is primarily a technical paper, which combines big data style of analysis with complex image analysis, and is therefore very interesting for the field. For this part, however, I am unable to assess the accuracy of the approach.

We thank the reviewer for his evaluation and we are pleased that the reviewer acknowledges the relevance of our work.

In terms of immunology, this study does not add much to the current knowledge, except that ILCs appear clustered close to plasma cells in tonsils, and that a subset of ILCs appear to express markers of plasma cells (the C4a cluster in Fig. 5). This latter claim is however weakly substantiated. The few markers used to define cluster C4a are not sufficient to claim that these cells are ILCs. Thus, the authors should do much more to verify this claim, for example by flow cytometry.

We agree that the claims on the identification of novel ILC markers had to be strengthened by further experiments. To validate our previous histology data that pointed to a subpopulation of ILCs expressing CD138 and IRF4, we included microarray data from ILC3 and 2 different populations of CD34+ hematopoietic progenitor cells for ILCs (as control population) isolated from human tonsil, as well as flow cytometry data in the revised version of the manuscript (Fig. 4 and Fig. S4). These data support our previous findings generated by multiplexed histology. In the case of IRF4, we were able to generate robust data confirming the expression of this transcription factor in tonsillar ILC3s. To our knowledge, this is the first report, which confirms IRF4 expression not only at the transcriptional, but also the protein level, in ILCs. It establishes this transcription factor as a marker for the identification of a subpopulation of ILCs by histology and flow cytometry. This not only underlines the relevance of our method for the identification of new markers even in rare cell populations, but, beyond this technological impact, will serve to reveal novel aspects of ILC biology.

Regarding CD138, we were able to confirm expression in ILCs on a transcriptional level. However, validation of the data by flow cytometry was more difficult compared to IRF4. This may partly be due to technical reasons. As discussed in the manuscript, CD138 staining in flow cytometry has been reported to be technically challenging, even when fresh tissue is used^{2,3}. Due to the ongoing COVID-19 pandemic during the revision of this manuscript, our access to fresh tonsil tissue was very restricted, as all elective surgeries (such as tonsillectomies) have been cancelled at the surgery wards we obtain them from. We therefore had to resort to frozen cells from tonsils, which may have impacted on the quality of the CD138 staining. We discussed both IRF4 and CD138 in the manuscript, but phrased our conclusions on the latter more carefully, in order to account the issues described above. Furthermore, we here for the first time define characteristic stromal microenvironments for human ILCs, that we find to be conserved across several tissues. For those reasons, we believe that this work adds substantial knowledge to the field of ILC biology, beyond the technical aspects of

the paper. We are confident that this study paves the way for future histological studies of human ILCs, which, until now, have been limited.

Reviewer #2 (Imaging immune systems, signaling) (Remarks to the Author):

The authors present a multiplexed fluorescence technology based on cycles of staining and bleaching with mostly PE labeled antibodies, imaging, image registration, segmentation, measurement and analysis of single cells. The target of the analysis is “helper” ILC, a rare population overall, and an important subset to localize in human tissues requiring multiplexing. As a rare population, ILC are expected to be surrounded by other cells. The pipeline seems robust and is open source so potentially accessible to a wider range of scientists to adopt on their own without committing to an expensive commercial platform. I would have appreciated a more explicit discussion of how the analysis pipeline copes with the obvious problem that due to limitations of resolution and membrane to membrane tiling of cells that cells are contaminated with signals from adjacent cells. I have some concerns the authors can address.

We thank the reviewer for his recognition of our approach in terms of robustness and versatility. We are well aware of the limits in resolution of our microscopic system and the accurateness of the segmentation, and have added text in the discussion.

There are two aspects of the results that give me some concerns about an influence of possible cross-contamination on the system. First, they comment that ILC are clustered. I would expect this if there was a bias against detection of single ILC surrounded by lineage positive cells.

We apologize if our phrasing was misleading. We do observe single ILCs in all tissues/samples analyzed, as shown for example in Fig. 3 B, D, H I and in Fig. 4 D, F. Our intention was to point out that, despite their rare abundance, ILCs are not only found as scattered single cells, but rather tend to localize in groups, suggesting their association with particular microenvironments, which lead to their preferred accumulation. We changed the text in order to stress this point.

Second, a couple of cases are raised where ILC in different locations share markers characteristic of their microenvironment. This was particularly surprising for the plasma cell marker CD138. The authors defend this result by saying that other B cell markers like Ig were not seen in the same ILC that are near plasma cells, but I think some independent validation that CD138 on ILC is needed- which could be done by multiparameter flow cytometry experiment or scRNAseq. My feeling is that the authors, and anyone trying to generate single cell datasets from immunofluorescence histology, needs to recognize this problem, quantify it as best they can, and then take advantage of it rather they trying to bury it. If the authors accept that the CD138 may be cross-contamination from neighboring plasma cells that is not getting filtered out and even verify that this is true, then they can use the contamination as an independent way to assess possible cell-cell contacts in the dataset. Looking at the data details presented it also looks like some membrane proteins have strong intracellular signals, maybe due to biosynthesis, turnover or storage, but these will be easier to assign to cells than proteins that are mostly at the plasma membrane. The IRF4 signal in the nucleus of some cells with associated CD127 signals is also clear. But validation of the unexpected combinations with a different method would build confidence in the novel conclusions.

We highly appreciate the expert comments and concerns on signal cross-contamination. Therefore, we included in the revised version of the manuscript transcriptome data obtained of two different CD34+ hematopoietic progenitor cell populations of ILCs, as well as mature ILC3 isolated from human tonsils, to

validate our findings (see below). Importantly, we understand that cross-contamination, although not applying to the main findings of the manuscript, is an issue in single-cell analysis of immunofluorescence histology data and that is why we added a paragraph on that topic in the revised manuscript.

One validation idea that might address a number of issues- take a single cell suspension of tonsil cells, attach them to PLL coated glass as low density (all cells well separated) and high density (all cells overlapping) and then run the pipeline. Does the analysis come out the same for analysis of the same number of cells well separated vs crowded at tissue density. I think this would be a good single cell ground truth and assessment of cross contamination due to shared cells borders. Perhaps this is better than flow cytometry as it allows use of same reagents and really tests the pipeline.

We thank the reviewer for his suggestion. We carefully considered performing this experiment. However, finally we decided to not follow the suggestion, after having performed validation experiments using two independent techniques. Please let us explain our decision:

In the case of IRF-4, our data are now supported not only on the protein level, but also by transcriptome analyses. A large fraction (~60%) of ILCs expressed this transcription factor in three separate flow cytometry experiments, and transcriptomic analyses also yielded robust statistic significance when the expression of IRF-4 was compared to hematopoietic progenitor populations from the same tissue. In addition, we found the results to be in line with published RNA-sequencing data cite, which showed elevated levels of the transcript in ILC3, compared to ILC1, ILC2 and NK cells.

For CD138, the situation is less clear. Although we also found significantly higher expression of the transcript in ILC3 compared to hematopoietic progenitors, suggesting its association with mature, tissue-resident ILCs, its validation by flow cytometry was challenging. As discussed in the text, the detection of CD138 by flow cytometry has been reported to yield negative results even in plasma cells, which are known to express this glycoprotein at high levels ^{2,3}. In addition, as laid out above, our access to fresh tonsil tissue was very restricted, as all elective surgeries (such as tonsillectomies) have been canceled at the surgery wards we obtain them from, due to the COVID-19 pandemic. We therefore had to resort to frozen cells from tonsils, which may have impacted on the quality of the CD138 staining. For the same reason, we could not obtain fresh material for the experiment suggested by the reviewer. Finally, a successful realization of the experiment suggested by the reviewer would be very challenging, given the rare abundance of ILCs in tonsils (0.15 to 0.50 %, as shown in the manuscript). Even in times of unrestricted access to tissues, we believe that setting up a system to detect ILCs on a glass slide would be beyond the scope of this manuscript. We hope for the understanding of the reviewer regarding this point.

Reviewer #3 (Computational analyses, systems biology) (Remarks to the Author):

The authors present a study using Multi Epitope Ligand Cartography (MELC) images of the human tonsil. This highly multiplexed imaging technique allows them to identify a number of cell types and to localize them in the tissue. The authors then focus on the analysis of the spatial distribution of Innate Lymphoid Cells (ILCs), which they address with two computational approaches with varying degrees of supervision.

Understanding tissue composition by the analysis of cell types in their spatial context is an important challenge today, for which a variety of imaging and analysis techniques exist, namely the many variants of multiplexed FISH or imaging mass cytometry. MELC is another technique to overcome the limitation in the number of simultaneously imaged markers. Proposed in 2006, MELC might nicely complement the arsenal of tools in this

regard. The application of MELC to analyze ILCs is certainly a novel application, even though the multiplexed analysis of tonsil sections (imaged with a different technique) has been published before.

In principle, I found the article interesting and timely. At the methodological level (which is the expertise I can contribute to the review process), I did not find much novelty: the authors use standard tools to perform image processing and analysis (background correction, pixel classification, object segmentation from the probability maps). A very similar version of the overall workflow has actually been proposed previously, even for the analysis of multiplexed images of tonsil sections, albeit for a different imaging technique [1]. On the other hand, the proposed method is solid (with some caveats mentioned below) and clearly serves the biological conclusions.

We thank the reviewer for this positive comment. We are pleased that the potential of our method to answer biological questions is recognized by the reviewer, and that he considers our method to be solid. We thank the reviewer for making us aware of the relevant publication, which we referenced in the manuscript.

The article in its present form has the following (potential) weaknesses:

- It is not very clear to me what the major contribution of the article is (biology, imaging or analysis). If the main result is on the imaging side, a deeper discussion of the advantages and drawbacks of MELC for this kind of application would be very helpful for the community. Regarding the analysis part: the computational aspects are handled correctly (with the exceptions mentioned below), but are not by themselves sufficiently new to justify publication. The novelty with respect to [1] seems limited to me.

We believe that the major contribution of our work is the novel application of a multiplex histology technique (MELC) that produces high-resolution images, to address questions in the biology of ILCs, an immune subset of great interest, that has been difficult to analyze in the tissue context until now. We combined our image acquisition with a computational analysis pipeline that allows us for the first time to identify and characterize ILCs and their microenvironments across several human tissues. Since ILCs are very rare compared to other immune subsets, their precise localization is largely unknown. Multiplexed techniques are required for their identification, so they conform a subset particularly challenging to study in histology. Moreover, the tissue-resident nature of ILCs calls for their analysis in the tissue context. In addition to the proper in situ identification of ILCs, we gained novel and potentially relevant information about ILC phenotype and localization within tissues that we plan to further investigate in future functional studies.

-The distinction between “hypothesis-driven” and “data-driven” in this context seems misleading to me. In the first approach, cell types are identified by rules formulated from the literature (i.e. expression of cell type specific marker genes). In the second approach, clustering is used to first identify all cell types, and then to characterize them according to rules from the literature (by inspection of the heatmap). In both approaches, prior knowledge is thus used to make sense of the data, in one case in an entirely supervised way, in the second case by cluster interpretation. Concretely, what would be the hypothesis used in the first approach, which is not used in the second? Also, I do not really understand what the second approach brings in addition to the first. If all cell types are known, it seems to me that the problem is effectively supervised and not unsupervised. If there was a new cell type or if the clustering was really essential to draw biological conclusions, this would be much more convincing. I elaborate on this below.

We thank the reviewer for his comment. It made us realize that we used a misleading terminology in the first submission. In this revised version, we eliminated all of the misleading terminology in the revised version of the manuscript.

- The analysis of the ILC niches is in principle the most interesting part of the article, but here the authors should stratify the niches according to the region where they are situated instead of comparing them to an overall statistic (see comment below).

We understand the point the reviewer raises and changed the text in the result and the discussion to underline the reasoning for our spatial analysis approaches. Please see below for a more detailed comment.

Other than that, I think the article illustrates very nicely what can be gained from spatial analysis. Also, the image data is of very high quality, and this sets this work apart from some of the previous studies, where only low-resolution images could be obtained.

We appreciate this positive feedback from an expert in the field.

Thomas Walter.

*[1] (1) Durand, M.; Walter, T.; Pirnay, T.; Naessens, T.; Gueguen, P.; Goudot, C.; Lameiras, S.; Chang, Q.; Talaei, N.; Ornatsky, O.; Vassilevskaia, T.; Baulande, S.; Amigorena, S.; Segura, E. Human Lymphoid Organ CDC2 and Macrophages Play Complementary Roles in T Follicular Helper Responses. *The Journal of Experimental Medicine* 2019, 216 (7), 1561 LP – 1581. <https://doi.org/10.1084/jem.20181994>.*

Detailed comments:

1. Normalization procedure: the authors propose to stretch the dynamic range of each individual image (page 5). This means that low intensity channels (for instance due to low expression) will be multiplied by a much larger factor than high intensity channels, which in turn can have substantial impact on the clustering procedure. The authors should motivate their choice, explain potential caveats and also provide some data to get a feeling for the impact on the clustering result.

We thank the reviewer for bringing up this important issue. Our initial motivation for stretching the dynamic range was to facilitate the visual representation and inspection of the data. We want to point out, that one strength of our analysis pipeline is its robustness, which makes it perform very similarly over a wide range of data pre-processing strategies. We have added two examples below, showing the t-SNE maps for the tonsil data (left, compare to Figure 4A of the manuscript) and colon data (right, compare to Figure 5B of the manuscript) without image normalization. For that reason, we think that the caveats are minimized.

In addition, we are currently working on basing our analysis not on mere signal intensities, but rather on signal-to-noise ratios, which will be an even more objective way to quantify signal strength. However, this is an ongoing project beyond the scope of this manuscript.

2. *Otsu-threshold: Random Forests provide posterior probabilities as output, i.e. the logical threshold would be 0.5 for a binary segmentation problem, or a simple max for a multi-class problem. It is not clear to me, why the authors would apply an Otsu-threshold on the posterior probability map.*

This procedure actually evolved during the refinement of our analysis pipeline. In that sense, Otsu thresholding is a remnant of our old pipeline, which used fluorescence images for nuclei and membranes, to perform thresholding-based segmentation instead of probability maps. Afterwards, we moved to probability maps to increase segmentation accuracy and since Otsu still performed well (as validated by visual inspection), we didn't change it. The Otsu thresholding actually proved to work as robust on probability maps as on fluorescence images. Since we had a robust solution, we had no motivation to change it, even if other threshold approaches are much faster. We have already followed the reviewer's advice for follow-up projects and implemented a simple manual thresholding procedure for binary segmentation.

3. *The procedure of assessing whether a cell is positive with respect to a particular marker was not entirely clear to me. The authors write "Subsequently, negative and positive cell populations for each marker were identified by intensity thresholding in order to classify cell-types, similar to conventional gating strategies performed in flow cytometry analysis." How did the authors set these thresholds?*

(e.g. by visual inspection of the images, or visual inspection of the histograms, etc.)

The thresholding was performed by visual inspection of the images. We added this information in the text of the revised manuscript.

4. *"Thus, ILCs were automatically classified and annotated as Lin- CD45+CD127+ cells.". First, I do not understand what the authors mean by "classify and annotate" (several occurrences in the text). Do they mean that the rule set Lin- CD45+CD127+ defines candidates that are then manually annotated by an expert? Or do they simply mean that ILCs were defined as Lin- CD45+CD127+ cells (which corresponds to applying thresholds*

in the respective channels)? If manual annotation is involved, what additional information would be used by the human annotator?

We apologize for the misleading phrasing. We actually simply meant “defined” (by applying thresholds), and since we understand that our phrasing could be misleading, we changed the text accordingly.

5. Analysis of ILC niches: in principle, this is the most interesting part of the study, because the spatial configurations can only be analyzed with image data (unlike cell type clustering and cell type calling, which can be done with molecular techniques). However, there is an issue with the statistical test. The authors compare the occurrences of cell types in the niches with occurrences outside. The problem with this is that the overall spatial distribution of cell types is very heterogeneous and compartmentalized. If we want to test whether the niches have a particular composition, one would have to compare to the distribution in a similar region (e.g. B-cell-region, crypt, T-cell-region, etc.).

Again, this was phrased misleadingly in the original submission, and we apologize. We have now changed the text referring to the spatial analysis of ILCs, in order to emphasize that two different spatial analysis approaches pointed to the same conclusion. The first approach was an ILC-centered neighborhood analysis, whereas the second one was starting from a broader tissue perspective. Both of them revealed ILCs and plasma cells to sit very close together in microenvironmental areas enriched with a defined stromal composition. We changed the text accordingly and hope that our point is now more clear.

6. “This tonsil region constitutes the tissue compartment that showed greatest statistically significant differences in terms of ILC accumulation, with p values ranging from 0.05 to 0.005, compared to the other tissue areas analyzed.” – The sentence is not very clear to me. Do the authors want to say that p-values of ILC accumulation vary most in the subepithelial connective tissue? What is the biological conclusion from this result?

We thank the reviewer for pointing out this confusing statement in the previous version of the manuscript. We aimed to emphasize that ILCs preferentially localize in the subepithelial connective tissue area and that the statistically significant differences for ILC localization are highest in this area, compared to the other defined tonsil areas. We changed the text and hope that the way it is phrased *now clearly makes the point*.

7. Clustering approach:

a. Like all clustering approaches, the final results are heavily impacted by the normalization procedure and correlation structure of the data. Often, we have to accept these shortcomings, because we are in a truly unsupervised setting, but this is not the case here: the authors perfectly know which combination of expression values defines the cell types. The limitation of these unsupervised approaches should at least be mentioned. For instance, I think that the number of 4 clusters is actually one among several possible solutions and might just represent the correlation structure of the chosen markers.

We thank the reviewer for his comment, which made us realize that we should present our results in a slightly different way. We agree that our setting is not truly unsupervised, and adapted the text and the figures accordingly.

b. The authors analyze the cluster distribution of ILCs. I would find it much more convincing to isolate all ILCs (supervised approach) and to provide hierarchical clustering using all markers for this subset of cells. This would have been a much more direct approach.

Following the advice of the reviewer, we now isolate all pre-classified ILCs and inspect their marker profile without subclustering this rather small population of cells. In the new version of the manuscript, we decided to focus on novel markers for ILCs, rather than a broad characterization and application of already known features.

c. In principle, there is nothing wrong with the analysis itself, but I somehow feel that it is not presented in the right light. The method is not truly unsupervised, as the markers are chosen and the clusters are interpreted according to prior knowledge; it is merely a way of grouping the data so that we can better impose our prior knowledge without extensive cell annotation. For instance, the authors state that “the data-driven approach presented here is suited for the identification and characterization of rare ILCs in several human tissues”, but this could have been achieved with their supervised approach.

We thank the reviewer for this important remark, and we agree that our presentation was somehow clumsy. We understand the importance of the point the reviewer raises and agree that the method used is not truly unsupervised, since we make use of a selected and limited antibody panel, through which we characterize cell populations in tissues. That is why we left out all misleading expressions, in the revised version of the manuscript. However, we would like to point out that we used the expressions “unsupervised” and “data-driven” to emphasize the fact that cell populations in this case are not pre-defined by intensity thresholding of images (and/or positivity/negativity of particular markers) in advance (as done by our first supervised or hypothesis-driven approach) but rather annotated a posteriori, once cells have been found to belong together and a part from the other cells, based on similarity distances calculated by clustering algorithms for all the markers contained in the panel.

We also agree with the reviewer with the fact that we could indeed identify ILCs with the *supervised* approach as well (by which we classified populations by intensity thresholding of a reduced combination of markers). However, with the *supervised/hypothesis-driven* approach, which we call *cell classification by image thresholding* in the revised version, we have to visually inspect the fluorescence images for all relevant markers in order to set thresholds for positivity/negativity and so classify cells. We understand that each manually set threshold implies a user bias and a certain degree of uncertainty while setting the threshold and it is additionally time-consuming. On top of that, intensity distributions in biological samples for one marker might vary in each independent experiment. Therefore, the thresholds set for one data set might not be extrapolated to others, making the comparison difficult. That is why, although we started performing *cell classification by image thresholding*, we were tempted to test the *less biased approaches*, which we call in the revised version *cell classification by clustering analysis*. It worked robustly in the several independent data sets analyzed and in different tissues and the grouping of cell populations was comparable to the groupings performed by fluorescence thresholding, with the advantage of being a less subjective, quicker, and straight-forward approach. Additionally, and probably most importantly, the clustering approach enabled the discovery of novel features among the ILCs that we would have most likely overseen if we had not performed this type of analysis. Hence, it has the potential to generate new hypotheses out of those imaging data.

Minor comments (typos, expression) :

- Page 6 : “Stroma-derived ECM proteins, such us fibronectin and smooth muscle actin (SMA) were also segmented and identified.”: the sentence is misleading (not the proteins are segmented).

- Page 6: “an area of ... was segmented”: technically, we would not speak of segmentation here, as the 10 μm neighbor region was determined without taking the image signal into account. I suggest to replace “segmented” by “defined”.

- “... and pixels on the other four data-sets were classified in an unsupervised manner” Actually, this is a misnomer. The segmentation method is supervised, and application of a trained classifier on new data does not make it unsupervised. The authors mean “without retraining”.

- “... and minimizes used-based variability in the analysis.” - I am not sure about this statement. Clearly, all automatic methods remove user-bias to some extent, and nobody would analyze these data completely manually. Among all automatic segmentation methods however, pixel-classification methods are not less user-biased (at the contrary). I would suggest to remove this half-sentence.

We thank the reviewer for making us aware of those issues. We changed the text accordingly.

References

1. Björklund, A. K. *et al.* The heterogeneity of human CD127+ innate lymphoid cells revealed by single-cell RNA sequencing. *Nat. Immunol.* **17**, 451–460 (2016).
2. Withers, D. R. *et al.* T cell-dependent survival of CD20+ and CD20- plasma cells in human secondary lymphoid tissue. *Blood* (2007). doi:10.1182/blood-2006-08-043414
3. Medina, F., Segundo, C., Campos-Caro, A., González-García, I. & Brieva, J. A. The heterogeneity shown by human plasma cells from tonsil, blood, and bone marrow reveals graded stages of increasing maturity, but local profiles of adhesion molecule expression. *Blood* (2002). doi:10.1182/blood.V99.6.2154

REVIEWER COMMENTS<

Reviewer #1 (Remarks to the Author):

The authors have attempted to address one on my main point concerning the identification of CD138, a plasma cell marker, as a new marker for ILC3s in tonsils. However their revision and rebuttal are not satisfactory. First, they claim that CD138 is difficult to stain in flow cytometry, while they obtain staining of this marker in histological section. This is surprising, as in general, flow cytometry stainings are more permissive to antibody staining if the samples are not fixed as in histology. Second, their definition of ILC3s in Fig 4G, as CD45+CD3-CD127+ (and not CD94/56 double positives) is rather poor. Their argument that ILC3s express CD138, which remains surprising and therefore to be clearly demonstrated, relies on the cytometry-based expression of IRF4, and then on the expression of CD138 transcripts in clusters of cells that express few markers, including IRF4 (based on the heatmap in 4D, showing only clustering of CD138 expression with CD38 and IRF4). Of note, the best marker for ILC3s is *Rorc* (in CD3 neg cells), which they do not mention. In addition, Figure 4A shows that the authors obtained quite few cells identified as ILCs. Thus, their findings on ILCs remain rather dubious.

Reviewer #2 (Remarks to the Author):

The authors have addressed my concerns. I was also very pleased to read the thoughtful response and revisions in response to reviewer 3. The solution that I proposed wouldn't have helped with the CD138 signal as the concern is related to proteolytic shedding during preparation of a single cell suspension, so the signal would likely still be lost whether flow cytometry or microscopy is used.

Reviewer #3 (Remarks to the Author):

The authors have addressed all the points I have raised in my previous review. I believe that this article makes a fine contribution to the field.

Point-by point response:

Reviewer #1 (Remarks to the Author):

The authors have attempted to address one on my main point concerning the identification of CD138, a plasma cell marker, as a new marker for ILC3s in tonsils. However their revision and rebuttal are not satisfactory.

We are sorry not to have been able to adequately address the concerns of the reviewer, and we have made an effort to do that in this revision. We want to point out that we agree with the reviewer that further studies are needed to clarify the expression of CD138 in ILCs, and that is why we removed the strong statements made in the previous version on the expression of this marker by ILCs in the main text. In addition, we clearly separated the findings on IRF4 from the findings on CD138 in the results part (page 8 and 9) and the discussion (page 13 and 14). We have also moved the transcriptional data on CD138 to supplementary materials (Fig. S5). Thereby, we aimed to express that the evidence supporting IRF4 expression in ILCs is certainly stronger than that of CD138, since histology, transcriptional analysis and flow cytometry all support IRF4 expression in ILCs, while in the case of CD138, only data using the two former methods point to an expression in ILCs. While we think that –particularly from the viewpoint of in situ analyses- the CD138 data are potentially interesting, we are now very carefully discussing the finding and its limitations, and have substantially toned down our conclusion on this subject, to avoid any dubious statements. We added a statement in the discussion, emphasizing that more analyses are needed to clarify this issue (page 14), however, as stated above, these analyses are currently impossible, and beyond the scope of this manuscript.

First, they claim that CD138 is difficult to stain in flow cytometry, while they obtain staining of this marker in histological section. This is surprising, as in general, flow cytometry stainings are more permissive to antibody staining if the samples are not fixed as in histology.

We agree with the reviewer that antibody stainings for surface markers in flow cytometry are more permissive, since cells are not fixed. However, it is also true that in some cases cellular receptors and other membrane proteins shed and/or internalize following the tissue disaggregation step needed to obtain single cell suspensions for flow cytometry studies. This has been previously shown particularly for CD138 (1,2) and we therefore discuss this fact in the manuscript (page 14). Nevertheless, we agree with the reviewer that we can only speculate about this being the cause of the discrepancies observed between the histology and

transcriptome data (which show CD138 expression on tonsillar ILC3) compared to the flow cytometry data (in which CD138 expression was absent on ILC3s). Consequently, we attempted to carefully phrase our results and discussion in the previous version of the manuscript regarding CD138 expression on tonsillar ILC3s. Nonetheless, in light of the concerns expressed by the reviewer and since we agree with him that our conclusions should be better supported, we further edited the main text in this revised version (page 7, 8, 9, 13, 14 and 15). We now clearly state that more experiments are needed to validate CD138 expression in ILCs.

Second, their definition of ILC3s in Fig 4G, as CD45⁺CD3⁻CD127⁺ (and not CD94/56 double positives) is rather poor.

We thank the reviewer for pointing out that our explanation of the flow cytometry data was poor and not clear enough. We have added a new paragraph in the results part of the revised version to clarify the gating strategy and the results shown in Fig. 4 G and Fig. S4, which belong together (page 8). We have also edited the figure legend (page 35 and 36).

In Fig. 4G, we first define helper ILCs as Lin⁻ (CD14⁻CD19⁻CD20⁻CD123⁻CD141⁻FcεR1α⁻) CD45⁺CD3⁻CD127⁺CD161⁺ cells. As the reviewer mentions from those we exclude CD56⁺CD94⁺ (double positive) cells, which we define as NK cells, in line with current literature (3). We then show that 98.6% of the remaining cells, defined as helper ILCs are c-Kit⁺CRTH2⁻ cells, which are consequently defined as ILC3s. We also depict that around 56.7% of helper ILCs express IRF4 and we show co-expression of this transcription factor with both c-Kit and NKp44 within the ILC population. Additionally, we show in Fig. S4 histograms of all lineage-defining transcription factors for the different ILC subpopulations (as defined in Fig. 4 G). In there, we show that the Rorγt⁺ fraction is composed of c-kit⁺CRTH2⁻ ILC3s (as defined in Fig. 4G). In Fig. S4 we also show the histogram of IRF4 (partly expressed by NKs and especially ILC3s) and of some ILC-relevant extracellular markers. Importantly, the ILC3 population is c-Kit⁺. The gating strategy and the ILC3 definition used in this manuscript have been broadly used previously in the literature (3, 4, 5, 6) and we added a sentence emphasizing this fact in the main text (page 8).

Their argument that ILC3s express CD138, which remains surprising and therefore to be clearly demonstrated, relies on the cytometry-based expression of IRF4, and then on the expression of CD138 transcripts in clusters of cells that express few markers, including IRF4 (based on the heatmap in 4D, showing only clustering of CD138 expression with CD38 and IRF4).

We understand the reviewer's concern on CD138 expression and although we tried to be cautious on the previous version when phrasing any conclusion on CD138 expression, we have now further edited the text on the revised version (see page 7, 8, 9, 13, 14 and 15), as mentioned above.

We argued that ILC3s potentially express CD138 based on the clustering analysis of the multiplexed histology data, which showed CD138 and IRF4 expression within the Lin⁻CD45⁺CD127⁺CD161⁺ cell cluster (Fig. 4 A-D). The heat-map in Fig 4D represents the relative expression level of 14 relevant markers on the protein level, analyzed by multiplexed histology, for every clustered ILC, shown to be Lin⁻CD45⁺CD127⁺CD161⁺ in Fig 4C. We confirmed CD138 and IRF4 expression by visual inspection of immunofluorescence images (Fig. 4 E). In the previous revision, we were asked to validate these findings by flow cytometry and/or transcriptome analyses, and we attempted to do both. Under the current circumstances, it was not easy for us to get human material to perform all the desired experiments and that is why we made use of transcriptome data from already published and well-characterized data of ILC3s that had been generated previously by some of the co-authors of this manuscript (3). The IRF4 and CD138 normalized transcript reads in sorted ILC3s and progenitor populations, which are now shown in two separate figures (IRF4 in Fig 4F, and CD138 in Fig S5), in order to separate the two findings and tone down the results on CD138 expression, were extracted from the aforementioned published study (3). Other transcript reads for the sorted tonsillar ILC3s, clearly demonstrating ILC3 identity, can be consulted in (3) and include Rorc and c-kit. Regarding the sorting strategy used, in short, we here cite from the supplemental information of the original publication: *'for microarray analysis, tonsillar ILC3s ... were sorted after MACS-depletion of CD34+ cells followed by CD56+ cell magnetic enrichment as DAPI-Lin-CD94-CD127^{hi}CD56+ ILC3s'*. The transcriptome data supported our hypothesis, which was based on clustering analysis of multiplexed histology data, on the expression of both IRF4 and CD138 in ILCs. In parallel, we performed flow cytometry analysis with the fresh tonsils we obtained, and with frozen material. However, we could only validate the protein expression of IRF4, but not of CD138 by flow cytometry. Since we understand that the robustness of the results shown are different between the two markers, we have now taken out all statements in the main text about CD138 being expressed by ILCs, and we also added a new sentence to emphasize that further studies would be required to validate the existence of CD138⁺ ILC3 (page 14).

We understand that it was confusing to show the results obtained by three different techniques with a similar visualization strategy (i.e. heat maps) in the same figure. We have added explanatory text in Figure 4D, and a sentence in the main text (page 8) to avoid any confusion in this regard.

Of note, the best marker for ILC3s is Rorc (in CD3 neg cells), which they do not mention.

We completely agree with the reviewer and we are aware of the importance of Rorc for ILC3 identification. Rorc was indeed shown to be expressed, as a proof of principle, on the DAPI⁻Lin⁻CD94⁻CD127^{hi}CD56⁺ sorted ILC3 population used for the microarray data (3). In addition, Rorγt was used to confirm ILC3 identity in the

flow cytometry analysis shown in the manuscript (Fig. S4). Consequently, we have now also added a sentence to the main text, in order to emphasize ROR γ t expression in the flow cytometry analysis of ILCs (page 8).

In immunofluorescence-based histology, we have successfully stained ROR γ t in human tonsils, and we could observe IRF4 and ROR γ t co-expression in the nuclei of tonsillar CD3⁻ cells in several cases. However, ROR γ t staining, as well as staining of some other transcription factors, is very much dependent on optimal tissue preparation and cryopreservation. When working with human material, we cannot always control these steps, since we receive the samples from various clinical collaborators. In addition, the dynamic range and sensitivity in immunofluorescence histology is lower than in flow cytometry. Together, this makes it difficult to always obtain robust and consistent results for ROR γ t, and we therefore decided to exclude it from the MELC analysis. Actually, the fact that MELC allows for an extensive combination of markers working reliably and robustly, is one of the advantages that make it particularly useful for the analysis of ILCs in histology (even without ROR γ t) and allows us to define ILC3 as Lin⁻CD45⁺CD127⁺CD161^{+/-}NKp44^{+/-}c-Kit⁺ cells, as previously published for flow cytometric analyses (3, 4, 5, 6).

However, in the data sets where we achieved ROR γ t staining, we indeed found co-expression of IRF4 and ROR γ t in CD3⁻ cells. We would like to show the reviewer one exemplary tonsil, where co-expression of IRF4 and ROR γ t can be observed on ILCs (localized around the B cell follicle and, therefore, confirming the localization pattern of ILCs shown in the manuscript):

Figure 4A shows that the authors obtained quite few cells identified as ILCs. Thus, their findings on ILCs remain rather dubious.

We agree with the reviewer that the number of ILCs is very low (10-30 ILCs/field of view in the 5 tonsils and 2 colon samples analyzed for this manuscript). However, the frequencies of ILCs in both tissues are in line with the literature (4,6). We understand that their low number is a limiting factor for the analysis of these cells in histology. While being aware of the fact that flow cytometry is the method of choice for the analysis of large cell numbers, in particular of rare cell populations, our approach is to provide a solution for the analysis of ILCs in the tissue context, which we believe can serve to address important questions in the field. The fact that we found robust localization patterns along and across tissues in relation to stromal landmarks makes us confident that we are detecting ILCs reliably. In addition, non-published data, which we generated in other human tissues point to the same direction, and we are further following this path. Furthermore, we confirmed our findings using two additional, high-throughput techniques, which analyze high cell numbers, as requested in the previous revision.

References

1. Medina, F., Segundo, C., Campos-Caro, A., González-García, I. & Brieva, J. A. The heterogeneity shown by human plasma cells from tonsil, blood, and bone marrow reveals graded stages of increasing maturity, but local profiles of adhesion molecule expression. *Blood* (2002). doi:10.1182/blood.V99.6.2154
2. Withers, D. R. *et al.* T cell-dependent survival of CD20+ and CD20- plasma cells in human secondary lymphoid tissue. *Blood* (2007). doi:10.1182/blood-2006-08-043414
3. Montaldo, E. *et al.* Human ROR γ t+CD34+ cells are lineage-specified progenitors of group 3 ROR γ t+ innate lymphoid cells. *Immunity* (2014). doi:10.1016/j.immuni.2014.11.010
4. Björklund, A. K. *et al.* The heterogeneity of human CD127+ innate lymphoid cells revealed

by single-cell RNA sequencing. *Nat. Immunol.* **17**, 451–460 (2016).

5. Lim AI, Di Santo JP. ILC-poiesis: Ensuring tissue ILC differentiation at the right place and time. *Eur J Immunol.* 2019 Jan;49(1):11-18. doi: 10.1002/eji.201747294. Epub 2018 Nov 2. PMID: 30350853.
6. Simoni, Y. & Newell, E. W. Dissecting human ILC heterogeneity: more than just three subsets. *Immunology* (2018). doi:10.1111/imm.12862

REVIEWERS' COMMENTS

Reviewer #1 (Remarks to the Author):

No more comments